# Objective and Algorithm Considerations When Optimizing the Number and Placement of Turbines in a Wind Power Plant

Andrew P. J. Stanley, Owen Roberts, Jennifer King, and Christopher J. Bay

National Renewable Energy Laboratory, National Wind Technology Center, Boulder, CO 80303 USA

**Correspondence:** Andrew P. J. Stanley (PJ.Stanley@nrel.gov)

**Abstract.** Optimizing turbine layout is a challenging problem that has been extensively researched in literature. However, optimizing the number of turbines within a given boundary has not been studied as extensively and is a difficult problem because it introduces discrete design variables and a discontinuous design space. An essential step in performing wind power plant layout optimization is to define the objective function, or value, that is used to express what is valuable to a wind power plant developer, such as annual energy production, cost of energy, or profit. In this paper, we demonstrate the importance of selecting the appropriate objective function when optimizing a wind power plant in a land-constrained site. We optimized several different wind power plants with different wind resources and boundary sizes. Results show that the optimal number of turbines varies drastically depending on the objective function. For a simple, one-dimensional, land-based scenario, we found that a wind power plant optimized for minimal cost of energy produced just 72% of the profit as the wind power plant optimized for maximum profit, which corresponded to a loss of about $2 million each year. This paper also compares the performance of several different optimization algorithms, including a novel repeated-sweep algorithm that we developed. We found that the performance of each algorithm depended on the number of design variables in the problem as well as the objective function.

## 1 Introduction

Wind energy provides several advantages to the sustainable energy grid of the future. Wind turbines produce minimal carbon dioxide or other air pollution, require no external fuel during operation, and require little water throughout their lifetime (Meldrum et al., 2013). Additionally, wind plants have an energy payback time of less than a year and can produce energy in an economically efficient manner (Razdan and Garrett, 2017; Vestas, 2020). In fact, wind energy has been a central focus of research and development in past decades such that, currently, wind is one of the cheapest sources of energy available (Lazard, 2018). Because of its many benefits, but in large part due to the economic drivers, wind energy installations have grown throughout the world as has the relative share of energy produced by wind. In coming years, wind energy technology is projected to improve, and wind energy capacity and market penetration are to increase even further (U.S. Energy Information Administration, 2019).

Because of economies of scale, utility-scale wind turbines are deployed in groups. This provides reduced cabling costs, easier construction and maintenance, and reduced land requirements. However, building turbines close together also introduces some challenges. One of these challenges is wake interaction between turbines. A wind turbine removes kinetic energy from

the air around it and converts this energy to electricity, creating a wake of slow-moving and turbulent wind behind it. When turbines are built close together, their wakes can reduce the amount of energy available in the wind, causing downstream turbines to produce less energy as a result. One way to reduce wake interactions between turbines is through wind power plant layout optimization. This optimization process involves determining the number of turbines to build in a wind power plant and their locations in order to reduce wake interactions and maximize performance. Finding an optimal wind power plant layout is a challenging, nonconvex problem with many interacting design variables. It is difficult to solve this problem without mathematical optimization tools because it often requires not-so-obvious trade-offs to reach a solution.

Appropriate methods of determining turbine locations within a wind power plant have been intensively studied, and researchers have demonstrated several methods that can be used to effectively optimize a wind power plant layout. The literature demonstrates a preference for gradient-free optimization methods applied to wind power plant layout optimization, and different studies showed success using genetic algorithms (Grady et al., 2005; Mittal, 2010; Abdelsalam and El-Shorbagy, 2018), greedy algorithms (Song et al., 2015; Chen et al., 2016), particle swarm methods (Pookpunt and Ongsakul, 2013; Hou et al., 2015), and random search (Feng and Shen, 2013, 2015) to determine improved wind power plant layouts (Hou et al., 2019). A common layout optimization method is to divide the wind power plant domain into a grid that defines possible turbine locations (at the center of the grid cells or at the intersections of the lines). One of the previously mentioned optimization methods is then used to determine at which of the predefined locations a turbine should be placed. In more recent years, some studies also showed success optimizing wind power plant layouts with gradient-based methods (Thomas and Ning, 2018; Stanley and Ning, 2019a; Baker et al., 2019). This type of optimization requires a continuous design space and computationally or analytically provided gradients that increase the complexity of the problem formulation. However, the computational expense required for gradient-based optimization scales favorably with increasing numbers of design variables compared to gradient-free methods for which the computational expense scales very poorly.

Layout optimization studies are almost always performed assuming that the number of turbines in the wind power plant is previously known. Determining the optimal number of turbines in a wind power plant is a much more difficult problem to solve because it requires the optimization of at least one integer design variable or a discontinuous design space. Although it has not been discussed in the literature as much as layout optimization determining turbine placement, optimizing the number of turbines in a wind power plant has also been researched in previous studies. Mosetti et al. (1994) first addressed this issue when they divided a wind power plant domain into 100 square cells as candidate turbine locations then used a genetic algorithm to determine the optimal number of turbines and at which of the potential locations they should be placed. Since this seminal paper was published, many other researchers proposed improvements to Mosetti's methodology and were able to find improved results, mostly by using new and better optimizers (Grady et al., 2005; Zergane et al., 2018; Ituarte-Villarreal and Espiritu, 2011; Moorthy and Deshmukh, 2015). Additionally, some applied a similar methodology to optimizing turbine number and layout at real geographical locations (Şişbot et al., 2010; Khanali et al., 2018). The vast majority of these more recent studies kept the same general optimization strategy, performed by dividing the wind power plant domain into a grid that defines potential turbine locations and using some optimizer to determine the best layouts.

60     Selecting the appropriate optimization methodology is a vital part of the wind power plant layout optimization process because it determines the quality of the final solution as well as the required computational expense. In addition to the optimization algorithm, a critical step is to appropriately select the objective function. For wind power plant layout optimization studies, objectives that are often considered are annual energy production (AEP) or cost of energy (COE). With a fixed number of turbines, the objective may not have much of an effect on the final solution. However, when the number of turbines is also

65 being optimized, the objective function can have a profound effect on the final optimized layout. For one set of optimizations discussed in Sec. 6.4, the optimal number of turbines ranges from 15–54, and the annual costs range from \$6.75 million to \$21.96 million, depending on if the plant was optimized for AEP, COE, or profit.

    For this paper, we studied two specific considerations in optimizing the number of turbines and their layout in a wind power plant. First, how different objective functions alter the optimized number of wind turbines and their layout in a wind power

70 plant. Li et al. (2017) began to explore this sensitivity with multi-objective optimization of wind farm layout and turbine number, considering AEP and COE. As part of their paper, these authors examined how different formulations of the COE definition affected the final solutions. Balasubramanian et al. (2020) also mention the importance of appropriately defining the objective for wind farm layout optimization. For our paper, we included an empirically based cost model, and compared three different objectives in our single objective optimization formulation to further understand the sensitivity of wind farm layout

75 and turbine number to the objective. Second, we tested using different problem formulations and optimization algorithms in finding a solution. In past research on wind farm layout optimization, there has been a wide variety of algorithms and problem formulations used, with little consensus on which strategies are the best (Shakoor et al., 2016; Baker et al., 2019; Hou et al., 2019; Balasubramanian et al., 2020). For this paper, we specifically studied how different algorithms performed depending on the objective and the size of the optimization problem. We compared a genetic algorithm and a greedy algorithm in a gridded

80 wind power plant domain, two commonly used wind power plant optimization methods, as well as a genetic algorithm with the boundary-grid method and a new repeated-sweep algorithm in a gridded domain.

    The rest of this paper is outlined as follows: Sec. 2 presents the wake model we used in this paper, and the relevant turbine parameters, Sec. 3 presents the power models, cost models, and how they are combined to form the 3 objective functions we explored in this paper, Sec. 4 describes the different sets of design variables we used to define the locations of wind

85 turbines, Sec. 5 explains the optimization algorithms we used in this paper, Sec. 6 presents and discusses the results from our optimizations, Sec. 7 explains the empirical considerations of this work and gives a general overview of the different scenarios we considered, and Sec. 8 contains our conclusions from this work.

## 2   Wake Model

The wind speed downstream of a turbine is reduced because turbines extract energy from the flow and from the complex

90 physics of the wakes they produce. In this paper, the desirability of the wind power plants we examined were dependent, to a large extent, on energy production. This energy production is a function of the wind speeds throughout the wind power plant. To calculate the wind speeds to be used in turbine power calculations, we used an analytic Gaussian wake model (Bastankhah

and Porté-Agel, 2016; Abkar and Porté-Agel, 2015; Niayifar and Porté-Agel, 2016). The wake calculations were performed using FLOw Redirection and Induction in Steady State (FLORIS), which is a computationally inexpensive, controls-oriented tool to calculate the steady-state flow field in a wind power plant (NREL, 2020). We include a brief description of the Gaussian wake model in this paper but, for more details, refer to the original model paper (Bastankhah and Porté-Agel, 2016).

Using the Gaussian wake model, the velocity of the wake behind a turbine is computed with the following analytical expressions:

$$
\frac{u(x,y,z)}{U_\infty} = 1 - C e^{-(y-\delta)^2/2\sigma_y^2 - (z-z_h)^2/2\sigma_z^2}
$$

$$
C = 1 - \sqrt{1 - \frac{D^2 C_T \cos(\gamma)}{8\sigma_y \sigma_z}}
$$

(1)

where $u$ is the velocity at a desired location $(x,y,z)$, where $x$, $y$, and $z$ refer to the streamwise, cross-stream, and vertical coordinates, respectively, $U_\infty$ is the freestream velocity, $C$ is the velocity deficit at the wake center, $y - \delta$ is the cross-stream distance between the point of interest and the wake center (where $\delta$ is the $y$ coordinate of the wake center and is assumed to extend straight back from the turbine creating the wake if the turbine is not yawed), $z - z_h$ is the vertical distance between the point of interest and the wake center (where the wake center is assumed to be at $z_h$, the hub height of the turbine creating the wake), $D$ is the rotor diameter, $C_T$ is the thrust coefficient, $\gamma$ is the rotor yaw angle (which is assumed to be 0 in this paper), $\sigma_y$ defines the wake width in the $y$ direction, and $\sigma_z$ defines the wake width in the $z$ direction. The distributions $\sigma_z$ and $\sigma_y$ are defined as:

$$
\frac{\sigma_z}{D} = k_z \frac{(x - x_0)}{D} + \frac{\sigma_{z0}}{D}
$$

$$
\frac{\sigma_y}{D} = k_y \frac{(x - x_0)}{D} + \frac{\sigma_{y0}}{D}
$$

where $x - x_0$ is the downstream distance between the point of interest $x$ and the onset of the far wake $x_0$, $\sigma_{y0}$ and $\sigma_{z0}$ refer to the wake width at the start of the far wake, $k_y$ defines the wake expansion in the lateral direction and $k_z$ defines the wake expansion in the vertical direction. The length of the near wake is defined as:

$$
\frac{x_0}{D} = \frac{\cos\gamma(1 + \sqrt{1 - C_T})}{\sqrt{2}[4\alpha I + 2\beta(1 - \sqrt{1 - C_T})]}
$$

where $\alpha = 0.58$, $\beta = 0.077$, and $I$ is the incoming streamwise turbulence intensity for which we used a freestream turbulence of 6% and the Crespo-Hernández model for wake added turbulence (Crespo and Hernández, 1996). The variables $\sigma_{y0}$ and $\sigma_{z0}$ are given in the following equations:

$$
\frac{\sigma_{z0}}{D} = \frac{1}{2} \sqrt{\frac{u_R}{U_\infty + u_0}}
$$

$$
\frac{\sigma_{y0}}{D} = \frac{\sigma_{z0}}{D} \cos\gamma
$$

where , $u_R$ and $u_0$ are defined with the thrust coefficient $C_T$ and the freestream wind speed $U_\infty$:

$$
\frac{u_R}{U_\infty} = \frac{C_T}{2(1 - \sqrt{1 - C_T})}
$$

$$
\frac{u_0}{U_\infty} = \sqrt{1 - C_T}
$$

For this study, $k_y$ and $k_z$ are set to be equal, meaning the wake expands at the same rate in the lateral and vertical directions. These wake spreading parameters $k_y$ and $k_z$ are defined as:

$$k_z = k_y = k_a I + k_b$$

where $k_a = 0.38$ and $k_b = 0.004$. In the case of interacting wakes, the wake deficits were combined using the traditional sum of squares method (Katić et al., 1986). Equation 1 defines the wind speed, $u$, at a single desired point. To determine the average rotor wind speed used to calculate the power production of a wind turbine, we averaged the wind speeds sampled at nine locations across the swept rotor area, which is the default in FLORIS. These nine sample locations are shown in Fig. 1.

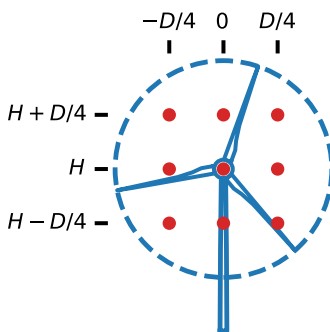

**Figure 1.** The nine locations at which the wind speeds are calculated across a wind turbine rotor. $D$ represents the rotor diameter, while $H$ is the turbine hub height. The effective turbine wind speed is determined as the average of the wind speed at each of these points.

For this study, we used a 2.5 -MW turbine definition. The turbine parameters are shown in Table 1, and the power and thrust coefficient curves, as well as the power curve, are shown in Fig. 2. As seen in the power curve, the rated wind speed is near 10 m/s.

**Table 1.** Wind turbine parameters.

| | |
|---|---|
| rated power | 2.5 MW |
| rotor diameter | 117.8 m |
| hub height | 88 m |

## 3 Objective Functions

For this paper, we explored three different objective functions in our wind power plant optimizations: 1) AEP, 2) COE, and 3) annual profit. In this section, each objective function is described in detail. We acknowledge that the models we used in this paper are simple. These simplified models are sufficient for this demonstration and investigation into varying results from different objectives; however, more detailed models can be easily included, depending on the use case and data available.

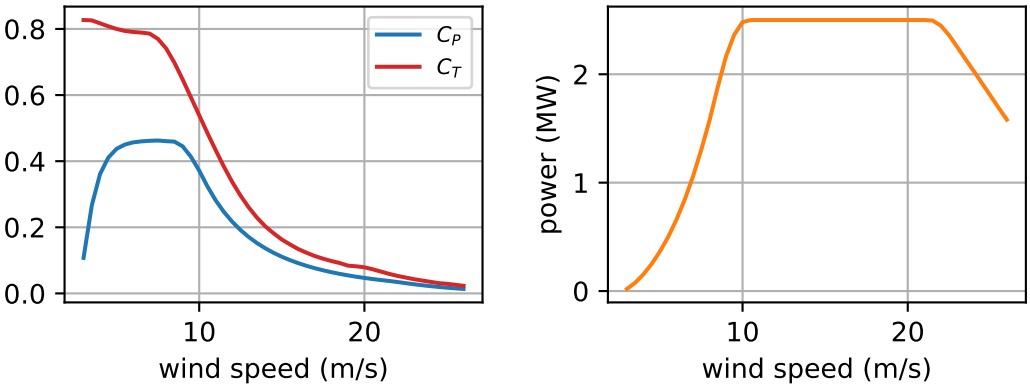

**Figure 2.** Left: $C_P$ and $C_T$ curves for the 2.5-MW turbine used in this study. Right: the power curve for this same turbine.

### 3.1 Annual Energy Production

AEP is a standard objective in wind power plant optimization (Pérez et al., 2013; Gebraad et al., 2017; Thomas and Ning, 2018). For problems where the value of energy produced by the wind power plant is fixed throughout its lifetime and independent of the time of day, and where the project cost remains constant or is not an important consideration, AEP is a reasonable objective. AEP optimization simply aims to maximize the energy production. For example, AEP is a common objective for wind power plant layout optimization where the turbine number and design are fixed. Typically to calculate AEP, the wind directions and

wind speeds are grouped into discrete bins in order to numerically calculate the integral:

$$\text{AEP} = 8,760 \sum_{i=1}^{n_d} \sum_{j=1}^{n_s} P_f(\phi_i, U(\phi_i)_j) f_i f_j$$

where 8,760 is the number of hours in a year, $n_d$ is the number of wind direction bins, $n_s$ is the number of wind speed bins per wind direction, $P_f$ is the power production of the wind power plant, $\phi$ is the wind direction, $U$ is the wind speed, and $f_i$ and $f_j$ are the frequency of wind associated with a given direction and speed.

The power of an individual turbine is calculated as follows:

$$P_t = \frac{1}{2} \rho A V^3 C_P(V)$$

where $P_t$ is the power produced by a single turbine; $\rho$ is the density of air, which we assumed is 1.225 $kg/m^3$; $A$ is the rotor swept area of the wind turbine; $C_P$ is the power coefficient of the turbine; and $V$ is the effective wind speed across the swept area, which for this paper was calculated with the wake model discussed in Sec. 2. When there is a variable number of turbines,

one can expect that a wind power plant optimized for maximum AEP will have many turbines spaced close together, filling the available land. If there are no penalties for costs considered in the optimization, additional turbines will lead to an improved objective, even if they are extremely inefficient and operate with high wake interference.

## 3.2 Cost of Energy

In some optimization problems, AEP may not be an appropriate objective as it does not account for the added cost or complexity
required to achieve gains in AEP. An example of this is wind turbine blade design, for which an increase in AEP comes at
the cost of additional mass and, therefore, cost. In a situation like this, it may be more appropriate to perform multi-objective
optimization or include both AEP and cost into a single objective. COE is another common metric used in wind power plant
design that captures both energy production and costs (Chen and MacDonald, 2014; Fleming et al., 2016; Stanley and Ning,
2019a). We calculated COE as a combination of costs divided by the AEP:

$$\text{COE} = \frac{\text{cost}}{\text{AEP}}$$

$$\text{cost} = \text{FCR}(\text{TCC} + \text{BOS}) + \text{O\&M}$$

where FCR is the annual fixed charge rate, which we assumed was 9.7% (Previsic, 2011); TCC is the turbine capital cost,
which we assumed is \$829 per kW of plant capacity (Wiser et al., 2020); O&M is the operation and maintenance cost, which
we assumed is \$44 per kW of plant capacity per year (Stehly and Beiter, 2020); and BOS is the balance of station cost. For
this paper, we created a simple relation of BOS costs as a function of the installed wind power plant capacity from a set of
higher fidelity BOS cost data (Key et al., 2020). This BOS cost function is shown in Fig. 3. As shown in the figure, the cost
per kW decreases as the total capacity increases because of economies of scale. One should expect that a wind power plant

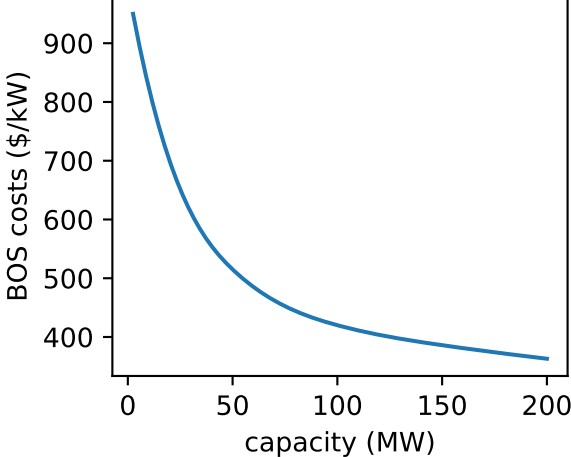

**Figure 3.** The balance of station (BOS) cost model (per kW of installed wind power plant capacity) as a function of the total power plant capacity (Key et al., 2020).

optimized for minimum COE would have fewer turbines than one optimized for AEP. This objective heavily considers the
170 additional costs from adding turbines to the wind plant. Extra turbines are only beneficial if the economies of scale from a cost
perspective outweigh the losses from additional wake interference that is introduced.

## 3.3 Annual Profit

Another metric that may be used for an objective function is annual profit. Like COE, this objective takes into account both energy production and costs. Additionally, an objective of profit can consider more refined measures of the value of energy, such as time-of-day pricing where the price of electricity varies depending on the time of day it is produced. Because a primary interest of most businesses is to make money, this objective would likely be of more interest to wind power plant developers, as opposed to AEP or COE previously discussed. For this paper, we defined profit simply with a fixed power purchase agreement as:

$$\text{profit} = \text{AEP} * \text{PPA} - \text{cost}$$

where PPA is the power purchase agreement, which determines the monetary value of the energy produced. We assumed that the PPA was a constant, as opposed to using time of day pricing, seasonal or yearly PPA adjustments, or including PPA incentives or penalties for power quality. For a given optimization the PPA was defined as a constant, although we varied this constant to study its effect during different optimizations. One should expect that a wind power plant optimized to maximize profit would have fewer turbines than one optimized for maximum AEP, but more turbines than one that is optimized for minimum COE. This objective still penalizes costs from adding more turbines, but may find solutions with slightly suboptimal COE as long as the AEP gains lead to sufficiently increased revenue.

## 4 Design Variable Parameterizations

For this paper, in addition to the different objective functions, we explored different optimization techniques and how they affect the final solution and the computational expense required to find it. One important part of any optimization is how to parameterize the design variables. In this section, we explain the two different parameterization methods we used in this paper: a gridded domain, where the number of design variables increases as the grid refinement squared; and a boundary-grid method, where the number of design variables remains constant at 11, regardless of the size of the domain or the number of turbines.

### 4.1 Gridded Domain Design Variables

The first set of design variables that we used in our optimization were similar to those initially used by Mosetti et al. (1994) and involved dividing the domain into a square grid of potential turbine locations. In this problem formulation, each of the grid points is a design variable, with the possible integer value of 1 (meaning a turbine exists in the associated position) or 0 (meaning the associated position is empty). Figure 4 shows this gridded domain for a square boundary with 8 row and column grid discretizations. Each of the blue points represents a design variable and is a potential location for a wind turbine. The computational expense required to optimize a problem generally scales poorly as the number of design variables increases. So, the grid must be refined enough to sufficiently search the design space, but not so refined that the optimization becomes computationally infeasible. Note that the number of design variables increases as the grid refinement squared, indicating that the number of design variables can quickly become impossible to optimize if the grid becomes too refined.

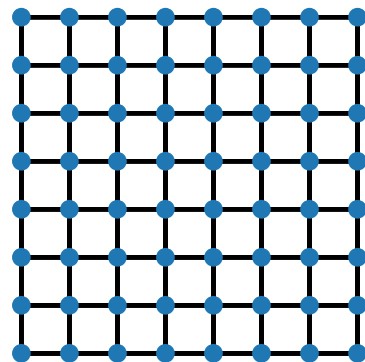

**Figure 4.** A square wind power plant that has been discretized with a square grid for wind turbine number and layout optimization.

## 4.2 Boundary-Grid Design Variables

The second parameterization that we optimized with was a modified version of the boundary-grid method. The boundary-grid parameterization is a simple method to define the layout of turbines in a wind power plant with very few design variables and still achieve layouts that perform just as well as wind power plants designed with more complex layout optimization techniques. In essence, it consists of placing some of the turbines around the boundary and the rest regularly arranged in a grid (Stanley and Ning, 2019b). The boundary-grid parameterization has the huge benefit of keeping the same number of design variables regardless of the number of turbines being optimized. This means that the layout of large wind power plants with hundreds or even thousands of wind turbines could be optimized without prohibitively high computational expense. In its original formulation, the boundary-grid method was defined for use with a gradient-based optimizer. This required the user to predefine some of the discrete variables that could not be optimized with gradients. Because we used gradient-free optimization in this paper, we slightly reformulated the boundary-grid method allowing integer variables and a discontinuous design space. In this paper, there are 11 design variables that describe the location of every turbine and are shown in Fig. 5.

The turbines in the interior of the wind power plant are arranged in a grid that is defined with 9 variables. First is the grid border, $B$, shown in Fig. 5a. The grid border is a number between 0 and 1 that defines the fraction of the boundary that will contain the inner grid turbines. When $B = 1$, the grid border is exactly the same as the wind power plant boundary, and proportionally decreases in size until $B = 0$, meaning the grid border vanishes in the center of the plant. The rest of the variables that describe the interior grid turbines define a complete distorted square grid of turbine locations; however, only those that are inside of the grid border are used. Figure 5b represents the other 8 variables used to define the locations of the interior grid turbines. The grid height and width are represented by $h$ and $w$, respectively. The center of the grid is shown as the point $(cx, cy)$, which determines the translation of all points in the grid, and the grid shear is shown in this figure as $\phi$. The

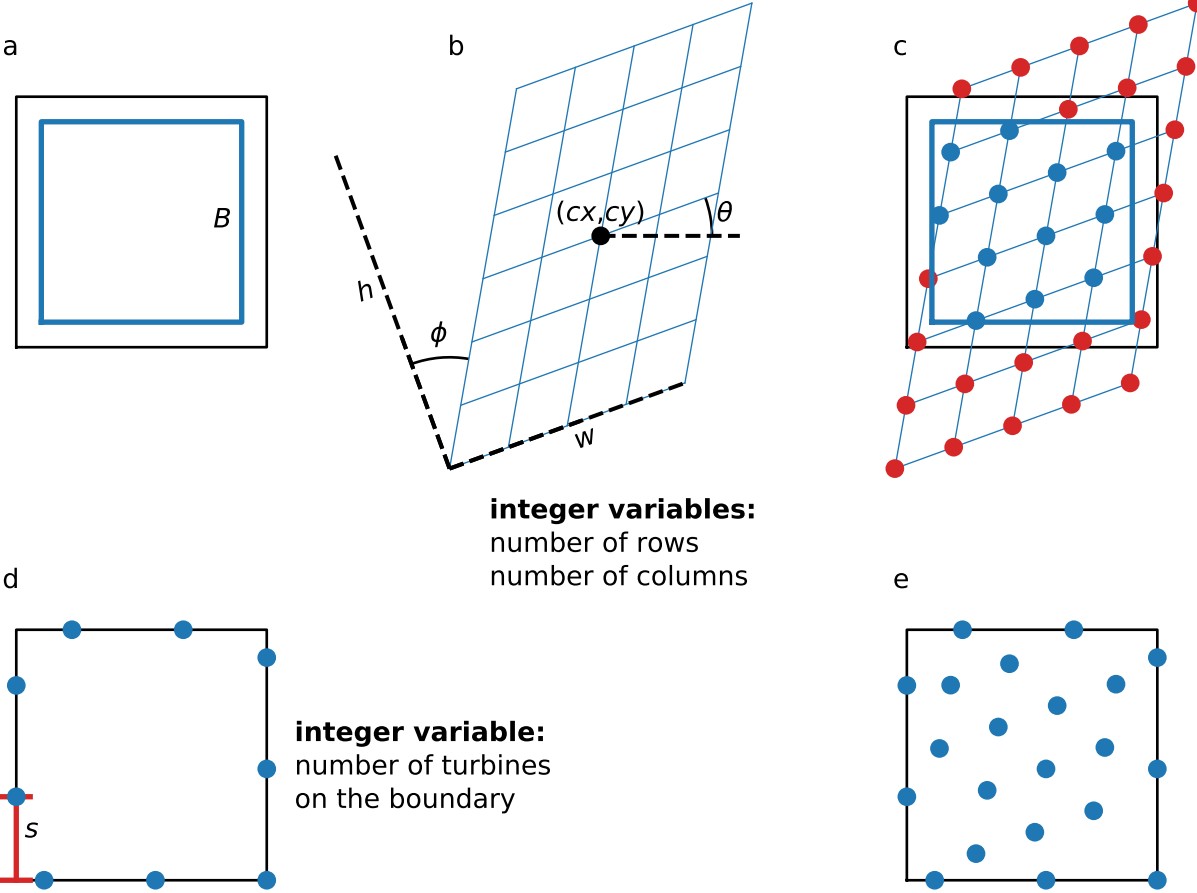

**Figure 5.** A description of the variables in the boundary-grid parameterization. 4a shows the grid border variable, $B$, and 4b shows the grid definition variables, $h$, $w$, $(cx, cy)$, $\phi$, and $\theta$, which represent the grid height, width, center, shear, and rotation, respectively. The number of grid rows and columns are also variables. 4c shows how all of the grid variables in 4a and 4b combine to define the interior grid turbine locations. 4d shows the boundary turbine variables, which are the boundary start location, $s$, and the number of boundary turbines. 4e shows the combined results of the boundary turbines and the interior grid turbines.

grid rotation (about the center $(cx, cy)$) is given by $\theta$. Finally, the number of rows and columns are also design variables used in the optimization. Figure 5c shows how the design variables in Figs. 5a and 5b are combined to obtain the turbine locations.

Turbines are placed at all of the grid intersection points that are inside of the grid border, shown by the blue dots. No turbines are placed at the grid intersection points outside of the grid border, indicated by the red dots in the figure.

The turbines around the boundary of the plant are equally spaced, traversing the perimeter of the plant. The boundary turbine locations are defined by two design variables represented in 5d. First, the number of turbines placed on the boundary is a design

variable. Second, the starting location of the first boundary turbine, represented as $s$ in Fig. 5d, is a design variable. The starting location is the distance from a constant anchor point at which the first boundary turbine is placed. Because the turbines are spaced equally around the wind power plant boundary, defining the location of this first boundary turbine implicitly defines the location of the rest of the boundary turbines. With the gridded domain design variables, the grid defines potential turbine locations which are assigned a Boolean value during the optimization to determine if they have a turbine. For the boundary-grid method, the design variables directly determine the location of every turbine in the farm, meaning there is always a turbine placed at the points defined by the boundary-grid parameterization. Figure 5e shows the final turbine locations defined by the boundary-grid parameterization variables shown in the rest of the figure. Notice that this is the combination of the boundary turbines in Fig. 5d and the blue inner grid turbines from Fig. 5c.

## 5   Optimization Algorithms

In this section, we discuss the details of the optimization algorithms we used in this paper. There are many algorithms that can be used to solve the wind power plant layout optimization problem, including determining the optimal number of wind turbines. For this paper, we chose to compare the performance of three gradient-free optimizers: a greedy algorithm, a genetic algorithm, and a novel repeated-sweep algorithm.

### 5.1   Greedy Algorithm

The first optimization algorithm that we used was a greedy algorithm. Several researchers in the past implemented a greedy algorithm in performing wind power plant layout optimization, making this a good benchmark (Changshui et al., 2011; Song et al., 2015; Chen et al., 2016). We applied our greedy algorithm to the gridded plant parameterization. For this algorithm, we started with one turbine placed in a random location within the plant domain. We then found the optimal location to place one additional turbine by evaluating the plant performance from placing the extra turbine at every potential turbine location in the grid. This process of adding one extra turbine was then repeated until adding an additional turbine did not cause an improvement in the objective. This algorithm is shown in Algorithm 1.

### 5.2   Genetic Algorithm

The second algorithm we used to optimize was a genetic algorithm. As with the greedy algorithm, genetic algorithms have also been a popular choice when performing wind power plant layout optimization studies with a discretized plant domain (Mosetti et al., 1994; Grady et al., 2005; Chen et al., 2013). We chose the tuning parameters for our algorithm with a combination of trial and error and best practice recommendations.

For the results shown in this paper, we performed single-point crossover and used a mutation rate of 2%. For the gridded plant domain, adjacent bits in the chromosome were adjacent in the plant domain. This helped create offspring that did not violate spacing constraints, as entire sections of the wind power plant that were traded during crossover would remain feasible (as long as they were feasible to begin with). After each generation, the entire population, consisting of parents and offspring,

**Algorithm 1** Greedy Algorithm

**Input:**

    number of bits: n

    objective function: obj

1:  initialize the best set of design variables A, an array of zeros of length n

2:  place a 1 in a random index of A

3:  initialize the best fitness value: $f = obj(A)$

4:  **while** not converged **do**

5:     set the iteration minimum to best fitness value: $m = f$

6:     **for** $i = 1 : \text{length}(n)$ **do**

7:         set temporary array B equal to A

8:         switch index i of B (from 0 to 1 or 1 to 0)

9:         evaluate the objective function with temporary array B: $g = obj(B)$

10:       **if** $g < m$ **then**

11:          redefine the iteration minimum: $m = g$

12:          define the iteration minimum array: $C = B$

13:       **end if**

14:     **end for**

15:     **if** $m = f$ **then**

16:       converged

17:     **else**

18:       redefine the best fitness value: $f = m$

19:       redefine the best set of design variables: $A = C$

20:     **end if**

21: **end while**

**Output:**

    best fitness value: f

    best set of design variables: A

were ranked in order of performance. The better-performing half of the entire population was kept to act as parents for the next generation. Convergence was assumed after the best performance was within a tolerance of $10^{-3}$ for 25 generations, or a maximum generation limit of 1000 was met. For the results in this paper, the maximum generation limit was never met. As the genetic algorithm was used for both the gridded parameterization and the boundary-grid method, continuous variables were binary encoded with 8 bits each. This means that for the boundary-grid parameterization, the variables were encoded into 76 bits—3 bits for the integer number of rows, 3 bits for the integer number of columns, 6 bits for the integer number of turbines on the boundary, and 8 bits for each of the 8 continuous variables. A rule of thumb for genetic algorithms is to use a population size of 10 times the number of design variables (Martins and Ning, 2020). For the gridded plant domain, we followed this rule of thumb exactly because there was a large number of design variables. For the boundary-grid parameterization, we used a population of 100, which was slightly less than 10 times the number of design variables. This gave us good results for our formulation. The genetic algorithm is represented is Algorithm 2.

### 5.3   Proposed Method: Repeated Sweep Algorithm

The last optimizer we used was a novel repeated sweep algorithm. As far as we are aware, no method similar to our proposed repeated sweep algorithm has been used in past research for wind power plant layout optimization. Like the greedy algorithm, this optimizer was only used with the gridded plant parameterization, where each of the design variables is an integer, either 0 or 1. The creation of this optimizer was inspired by attempting to apply gradient-based optimization principles to discrete design variables. As described below, the algorithm works by comparing adjacent points, and switching the values if it would improve the objective function, which could be imagined as the discrete version of a gradient. The repeated sweep algorithm consists of three phases. First is a single search phase, followed by two trade phases.

In the search phase, one by one and in a random order, the value at each potential turbine location is switched from 1 to 0 or from 0 to 1. If the objective improves, the swapped value is kept; if not, the design variable retains its original value. This is done until every potential turbine location has been evaluated, and the value has been changed or retained. In both trade phases, each potential turbine location is again searched through one by one. However, in these phases, instead of exploring adding or removing turbines (like in the search phase), the potential turbine location trades values with the cell adjacent to it. In the way we formulated the problem, in the first trade phase, each position trades places with the cell to the right; in the second trade phase, each position trades places with the cell above it. As with the search phase, if a trade results in an improvement in the objective, the trade is kept. If not, the trade is rejected and the original locations are retained. Also, as with the search phase, the trades are done in a random order until a trade at each location has been evaluated. The three phases are repeated in order, search-trade-trade, until the objective function does not improve after a complete cycle of all three phases. The repeated sweep algorithm is shown in Algorithm 3.

### 5.4   Gradient-Based Optimization

The optimization algorithms discussed previously are gradient-free, and can simultaneously optimize the number of turbines and their layout in a wind power plant. Another way to optimize turbine number and layout in a wind power plant is with

**Algorithm 2** Genetic Algorithm

**Input:**

    number of bits: `n`

    objective function: `obj`

    population size: `pop_size`

    maximum generations: `n_gens`

    crossover rate: `crossover`

    mutation rate: `mutation`

    tolerance: `tol`

    convergence generations: `conv_gens`

1: initialize the parent population `parent_pop`, a 2-D array of size `pop_size` by `n`

2: initialize the offspring population `offspring_pop`, the same size as `parent_pop`

3: initialize the generation counter, `generation` $= 1$

4: initialize the convergence criterion, `difference` $= 1$

5: **while** not converged **and** `generation` $<$ `n_gens` **do**

6:     perform crossover to create offspring population

7:     mutate (leave the best individual unmutated)

8:     evaluate fitness of offspring

9:     rank the fitness of the total population

10:     evaluate `difference`, |best fitness of current generation $-$ best fitness of generation `conv_gens` ago|

11:     **if** `difference` $\leq$ `tol` **then**

12:         converged

13:     **else**

14:         keep the best individuals between the current parent population and the offspring population as the parents for the next generation

15:     **end if**

16: **end while**

**Output:**

    best fitness value

    best design variables associated with the best fitness value

**Algorithm 3** Repeated-Sweep Algorithm

---

**Input:**

    number of rows and columns: `n, m`

    objective function: `obj`

---

1: randomly initialize a feasible array of design variables `A`, a 2-D array of size `n`-by-`m` with 1s and 0s. The 1s in the matrix correspond to the physical locations of turbines in the wind power plant. Ensure that this initialized matrix satisfies all of the constraints.

2: initialize the fitness: $f = \texttt{obj(A)}$

3: **while** not converged **do**

4:     cycle to the next step: search, trade rows, trade columns. Loop back to the search step after completing trade columns. We started with the search step.

5:     **for** sweep through all of the indices of `A` in a random order **do**

6:         set temporary array `B` equal to `A`

7:         **if** search step **then**

8:             change the value of the current index of `B` from 0 to 1 or 1 to 0

9:         **else if** trade rows step **then**

10:             trade the value of the current index of `B` with the value on its right

11:         **else if** trade columns step **then**

12:             trade the value of the current index of `B` with the value above it

13:         **end if**

14:         **if** $\texttt{obj(B)} < f$ **then**

15:             keep the switched matrix $A = B$, and update the fitness value $f = \texttt{obj(B)}$

16:         **end if**

17:     **end for**

18:     **if** three consecutive steps are completed (in any order) without improving the fitness value **then**

19:         converged

20:     **end if**

21: **end while**

---

**Output:**

    best fitness value: `f`

    best set of design variables: `A`

---

gradient-based optimization. Gradient-based algorithms cannot optimize integer design variables or discontinuous design spaces—both of which are conditions that apply to the problem addressed in this paper. However, it is possible to repeat a gradient-based optimization multiple times with different numbers of wind turbines, then choose the overall best solution for the given objective. This process is computationally expensive for two main reasons. First, *a priori*, it is difficult to determine the approximate number of turbines that will be optimal. This means it is necessary to repeat the optimization many times, using different numbers of wind turbines. Second, gradient-based optimizers are especially susceptible to converging to local minima in the design space. This problem is also prevalent in gradient-free optimization, but is more pronounced in gradient-based optimization. The problem can be mostly accounted for by repeating the optimization many times with different randomly initialized design variables, but this requires even more computation.

In this paper, our purpose was to compare some gradient-free methods that could be effectively used to solve for the optimal turbine number and placement in a wind power plant. For one case discussed in Sec. 6.2 we also used gradient-based optimization in order to compare the results. For the optimizations in this section, the time required to evaluate the objective function was small, allowing us to quickly perform the hundreds of optimizations necessary to explore the design space. To perform the gradient-based optimization, we swept through all of the possible numbers of wind turbines that could fit in the wind power plant without violating the spacing constraints, which was 2–18 turbines. For each number of turbines, we performed 50 optimizations with randomly initialized turbine locations. This gave us relatively high confidence that the solution we found with the gradient-based optimization was near the global solution. We used finite-difference gradients for these optimizations, which do not perform as well as analytic gradients, both in quality of the final solution and in computational expense. However, for the case in which we used the gradient-based optimizer, the wind rose was simple and the number of design variables was relatively small, meaning the finite-difference gradients performed sufficiently well. For the results in this paper, we used the open-source SLSQP (Sequential Least Squares Programming) optimizer available in SciPy (Virtanen et al., 2020).[1]

### 5.5 Constraints

In our layout optimizations, we assumed there were only two constraints—a spacing constraint and a boundary constraint. The turbines were constrained to be at least two rotor diameters apart from each other. This minimum spacing is on the small side, and is used to exaggerate the differences in the optimal solutions obtained with different objective function. The minimum spacing constraint implicitly defined the maximum number of turbines that could be placed in the wind power plant. Additionally, turbines were constrained to remain inside a prescribed boundary.

## 6   Results

In this section, we discuss the results of our wind power plant simulations and optimizations. Included in this section are a simple, one-dimensional (1D) sweep of the different objective functions versus the number of wind turbines, then full two-dimensional (2D) wind power plant layout optimizations run for the different objectives and with the different optimization

---

[1]https://docs.scipy.org/doc/scipy/reference/generated/scipy.optimize.minimize.html

algorithms. The wind plants that we optimized and discuss in this section are a small wind power plant with a unidirectional wind rose, a large wind power plant with a unidirectional wind rose, and a large wind power plant with a full wind rose. Finally in this section, we present results from optimizing wind power plants for maximum profit with varying PPAs.

## 6.1 1D Example

First we discuss a simple, 1D example to demonstrate the effect different objective functions have on the optimal solution. For this example, we simulated a single row of wind turbines in line with the wind, which had a constant speed of 10 m/s. The length of this row was fixed at 25 km, and the turbines were equally spaced. For this scenario, we calculated the value of each objective as a function of the number of wind turbines in the simple wind power plant; results are shown in Fig. 6.

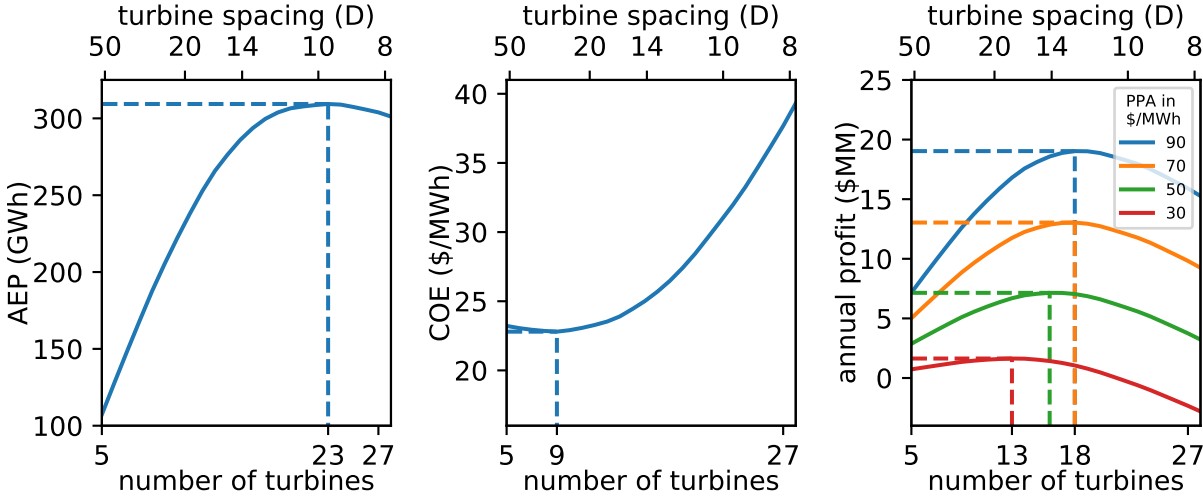

**Figure 6.** Different objectives as a function of the number of turbines in the wind power plant. For this example, the turbines are in line with the wind direction and are equally spaced in a wind power plant with a fixed length. From left to right, the objectives represented here are annual energy production (AEP), cost of energy (COE), and annual profit.

One key takeaway from this figure is that the optimal number of turbines for each objective is very different. Obviously, a wind plant designed for maximum AEP will look very different than wind plants designed with other objectives in mind. When maximizing AEP, there is no penalty for the extra costs associated with building extra turbines. As long as adding another turbine produces more energy, it is superior—no matter how marginal the increase in energy and how large the increase in cost. It makes sense that the wind power plant optimized for AEP has the most wind turbines, 23 in this example, because more turbines are added until the wake effect from adding an extra turbine outweighs the additional power it provides.

On the other hand, when COE is the objective, the optimal number of wind turbines is just nine, much lower than for the AEP objective. If the cost of the wind plant was modeled as a linear function of the number of turbines, the optimal COE solution

would be just one turbine. A single turbine would have no wake interference from other turbines and would, therefore, produce energy for the lowest cost. However, there are some economies of scale involved with wind plant development, represented in our cost model by the decrease in BOS costs with increasing power capacity. This means there is some optimum greater than one where the wake interference is still relatively low and the costs per turbine in the wind plant are decreasing steeply with additional turbines.

Finally, a completely different solution is obtained when optimizing the wind power plant for maximum profit. While the COE objective optimizes the ratio between the value a wind plant produces and the cost, the profit objective optimizes the difference between the value a wind plant produces and the cost. At first, it may seem nonintuitive that the solution for optimal profit is different than the solution for optimal COE because minimized costs should be related to more profit. In Fig. 6, notice that the optimal COE solution is 9 turbines and produces a COE of about $23/MWh. In this case, energy generation and, 350 therefore, revenue generation, are limited because of the small number of turbines. For 18 wind turbines, a slightly higher COE is achieved of about $25/MWh. From a COE perspective, this is suboptimal. However, the additional revenue produced from the extra turbines outweighs the increase in COE. The exact number of turbines for optimal profit depends on the monetary value of the energy, which is defined with the PPA. This means that the optimal solution is different depending on the PPA, represented by the different colors in the subfigure on the right. The optimal number of turbines increases from 13 to 16 as 355 the PPA increases from $30/MWh to $50/MWh, then again to 18 as the PPA increases to $70/MWh. However, when the PPA increases to $90/MWh, the optimal number of turbines remains at 18. This is because the number of turbines is not continuous, and is only represented by integer values. For a given scenario, different PPA thresholds could be defined, above which the optimal number of turbines would increase by one. From Fig. 6, it appears that the optimal number of turbines is more sensitive at low PPA values, and becomes less sensitive as PPA increases.

As described previously, Fig. 6 shows different metrics as a function of the number of turbines in a 1D wind power plant. In addition to the number of turbines shown on the bottom axis, this figure also shows the turbine spacing in rotor diameters on the top axis. While this axis is useful in understanding the results from a more familiar perspective, one must be careful not to interpret these values incorrectly. The optimal turbine rotor diameter spacing reported in this figure is around 10 for AEP, 24 for COE, and between 12 and 18 for profit. These are very large turbine spacings for a wind plant and are likely infeasible 365 because of land or cabling constraints. One must remember that these results were obtained from a very simple design space sweep with all wind turbines exactly in line with the wind. Because real wind power plants are built in two dimensions, with full wind-direction variability, it is actually possible to build turbines much closer together than is indicated in Fig. 6.

As demonstrated in Fig. 6, when determining the number of wind turbines to build in a wind power plant, an appropriate objective is essential to achieving a desirable solution. This is true of any optimization problem, but it is particularly important 370 to remember for this application. COE is an extremely common objective function in wind plant design—and rightfully so. However, as demonstrated in this simple example, the optimal number of turbines to minimize COE is suboptimal if the aim is to maximize annual profit, which may (or may not) be what is most important to those designing the wind power plant. In this specific example, the optimal number of turbines for COE results in $5.1 million of annual profit with a PPA of $50/MWh, just 72% of the optimal profit of $7.1 million. This significant difference in the optimal performance and wind plant design

for different objectives has important implications for techno-economic considerations in wind plant design. Economic factors drastically change the optimal solution, which highlights the importance of having accurate cost models, and again identifying the correct objective for design and optimization.

Historically, capacity expansion models have assumed a constant power density that does not vary with the PPA. Not only does Fig. 6 demonstrate the differences between a minimum COE objective and a maximum profit objective, but it also shows that the cost modeling assumptions can greatly affect the optimal number of turbines in a given land-constrained wind plant. Aggressive carbon reduction scenarios or other renewable energy goals typically result in high PPAs for renewables, which would lead to a higher optimal number of turbines and higher capacity densities for land constrained sites. This has important implications for capacity expansion models and could play a role in the future deployment of wind, as capacity density may often be much higher than is currently assumed.

## 6.2 Small Plant with Unidirectional Wind Rose

With the 1D sweep of the design space complete to provide some intuition about the different objective functions, we now discuss a simple layout optimization for a small wind power plant with a unidirectional wind rose. As stated before, we performed the optimization of each objective using a gridded domain, optimized with a greedy algorithm, a genetic algorithm, and a repeated sweep algorithm. We also optimized a boundary-grid layout parameterization with a genetic algorithm. Also, as mentioned before, for this small wind plant, we optimized the layouts using gradient-based optimization.

For this small wind power plant optimization, we assumed the domain was square with 800-m sides. The wind came from a direction of 300 degrees, or 30 degrees north of west. The wind speed was assumed constant at 10 m/s, which is close to the rated wind speed for our turbine model. The PPA was assumed to be $30/MWh, which is close to the COE solutions that were achieved, and is within the range of the PPAs of real wind farms from the past few years (see Fig. 16). For the gridded design variables, the domain was discretized into a 10-by-10 grid. We ran each optimization method five times to convergence because the final solution is dependent on the randomly initialized population or design space. Because each of the optimization algorithms has some stochastic qualities, with enough time and randomly initialized starts, each optimization method will potentially be able to find a very good solution. However, we believe that five optimizations for each is enough to give a good idea of their performance relative to each other for each of the objective functions.

Results for the small wind power plant optimizations with a unidirectional wind rose are shown in Table 2 and Fig. 7. Table 2 shows the optimization results and the computational expense associated with each optimization method and for each objective function. The first column shows the objective function, and the second column shows the optimization method. The third column gives the optimized number of turbines in the wind plant, the fourth column shows the average turbine spacing in rotor diameters associated with that number of turbines, the fifth column provides the best solution from the 5 optimizations. In this column, the best and worst solutions are indicated and bold in the table. The sixth column shows the best solution normalized by the best solution out of all of the optimization methods for the given objective. The seventh and eighth columns provide the total time required to run the 5 optimizations, in seconds and hours, respectively. Finally, the ninth column shows total number of calls to the wind farm evaluation, or function calls, required to run the 5 optimizations for each optimization method. The

separate, italicized bottom row for each objective in this table show the gradient-based optimization results. Figure 7 shows the flow field for the best layout for each objective function. These are the layouts corresponding to the indicated bold cells in Table 2. These flow fields show a horizontal slice of the wind power plant at the turbine hub height. The black lines represent the wind turbines, the red areas represent faster freestream wind speed, and the blue areas represent a slower waked wind speed. We did not include a color legend because we only wish to demonstrate qualitative information with this figure; therefore, exact values are not important for this purpose.

**Table 2.** Complete optimization results for the small wind power plant with unidirectional wind rose.

| objective | optimization | num. turbines | avg. spacing (D) | optimal value | normalized optimal value | time (s) | time (hr) | function calls |
|---|---|---|---|---|---|---|---|---|
| AEP (GWh) | greedy grid | 12 | 2.76 | **238 (worst)** | 0.792 | 15 | 0.004 | 2,267 |
| | genetic grid | 16 | 2.26 | 293 | 0.973 | 1,867 | 0.52 | 105,872 |
| | sweep grid | 15 | 2.36 | 270 | 0.897 | 4 | 0.001 | 204 |
| | genetic BG | 16 | 2.26 | **301 (best)** | 1.000 | 1,022 | 0.28 | 50,519 |
| | *GB* | *17* | *2.17* | *299* | *0.993* | *12,082* | *3.36* | *651,265* |
| COE ($/MWh) | greedy grid | 10 | 3.14 | **22.16 (worst)** | 1.015 | 11 | 0.003 | 1,860 |
| | genetic grid | 11 | 2.93 | **21.84 (best)** | 1.000 | 1,443 | 0.40 | 106,755 |
| | sweep grid | 10 | 3.14 | 21.88 | 1.002 | 8 | 0.002 | 656 |
| | genetic BG | 9 | 3.40 | 22.16 | 1.014 | 319 | 0.09 | 26,604 |
| | *GB* | *11* | *2.93* | *21.93* | *1.004* | *12,071* | *3.35* | *655,392* |
| annual profit ($MM) | greedy grid | 11 | 2.93 | 1.88 | 0.918 | 15 | 0.004 | 2,208 |
| | genetic grid | 12 | 2.76 | 1.99 | 0.971 | 1,306 | 0.36 | 95,687 |
| | sweep grid | 12 | 2.76 | **1.76 (worst)** | 0.860 | 5 | 0.001 | 341 |
| | genetic BG | 14 | 2.48 | 1.85 | 0.905 | 532 | 0.15 | 36,510 |
| | *GB* | *13* | *2.61* | **2.04 (best)** | *1.000* | *13,130* | *3.65* | *702,650* |

### 6.2.1  Small Power Plant with Unidirectional Wind Rose: Different Objectives

First, we will discuss the differences between the optimal solutions for the different objective functions. For optimal AEP, the best solution has as many turbines as the optimizer can fit into the wind power plant without violating spacing constraints. As can be seen in the top subplot of Fig. 7, the optimal layout has turbines that are spaced very close together. Wakes are strong in the flow field, which contains several turbines that are fully or partially waked. For this objective, it does not matter if some turbines are greatly affected by wakes as long as their energy contribution is positive. Now, in this case, the optimal solution had the maximum number of turbines as could fit into the boundary. However, from Fig. 6 we see that even for the AEP objective, there is a point where adding additional turbines actually becomes detrimental. We also see from Fig. 6 that this could occur at a relatively large turbine spacing, between 9-10 rotor diameters. For the 1D sweep, the turbines are all exactly in

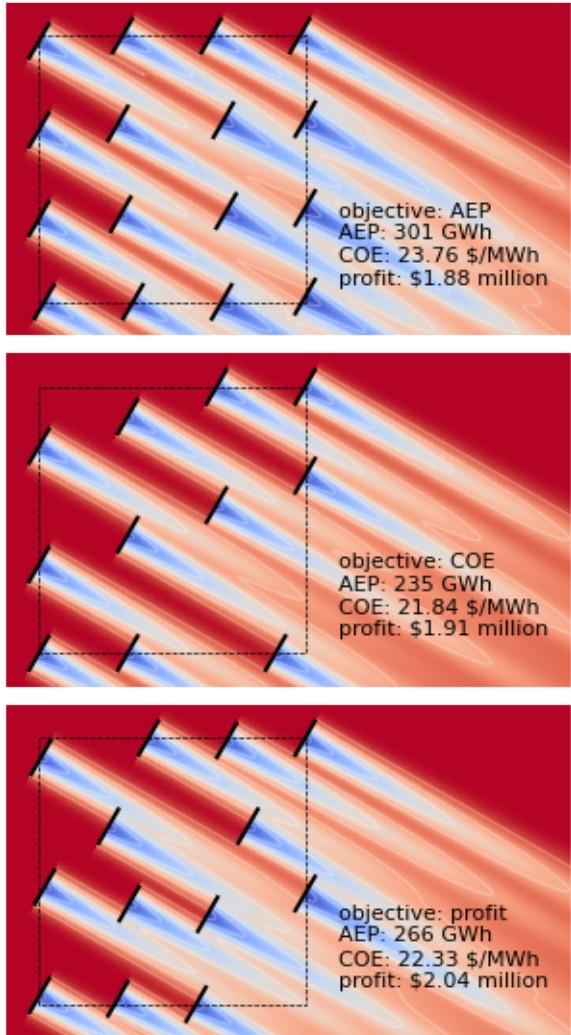

**Figure 7.** The optimal layouts for each objective for the small wind power plant with a unidirectional wind rose. From top to bottom, the associated objective functions are AEP, COE, and profit. The text within each figure provides the values for all three metrics for each wind plant.

line with the wind. Additionally, rather than having two or three turbines waked in line, there are many in line with each other. This indicates that, in large part, the AEP is determined by deep array effects. For the small wind plant layout optimization discussed in this section, there are at most three turbines in line with each other. In this case, adding turbines, even if they are fully waked, increases the AEP. If we were to repeat the optimization for a much larger domain we could potentially see results similar to Fig. 6, where having too many turbines could actually be detrimental for AEP.

While the wind power plant optimized for AEP maximizes the number of turbines in the design space, the wind plant optimized for minimum COE looks very different. This wind plant has 11 turbines, as opposed to 16 for maximum AEP. The turbines are arranged such that waking is minimal. For this objective, it appears that the optimizer maximizes the number of unwaked wind turbines. For this case, we can conclude that when the turbines are waked, the loss in energy production outweighs the benefits gained from economies of scale in the cost model. Therefore, additional turbines are good if they meet some minimum power production requirement, which is dictated by the cost model.

For the last objective, profit with a PPA of $30/MWh, the optimized number of turbines is 13. This is between the optimal number of turbines for the COE and AEP objectives. When optimizing for profit, the solution appears to be a balance between minimizing COE and maximizing AEP. A turbine is allowed to be waked as long as the gains from the energy it produces outweigh the costs of adding the extra turbine. As will be discussed in more detail in Sec. 6.6, the point where adding an additional turbine is no longer profitable is determined by the PPA. A lower PPA will drive the solution for maximum profit toward the solution for minimum COE, while a higher PPA will drive it toward the solution for maximum AEP.

We want to emphasize that the results we show are not meant to demonstrate exact solutions or guidelines for determining the number of turbines in a wind power plant. Specific solutions will depend on wind resources, turbine parameters, boundary shape and size, PPA, constraints, and other factors. Our purpose is to demonstrate that the optimal number of turbines and their layout are completely different depending on the objective. The true objective must be carefully formulated when optimizing the layout of a wind power plant. Figure 8 shows how the wind plants optimized for the three different objectives compare in other metrics. We demonstrated objectives of AEP, COE, and profit, and how they all produce different solutions. All of the wind plants optimized for a specific metric greatly underperform in the other metrics that we calculated. The one exception is the optimal profit solution, which also accomplishes a relatively low COE.

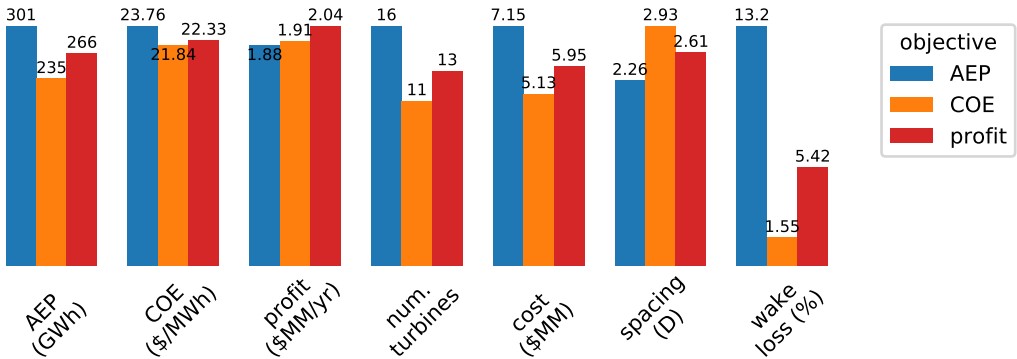

**Figure 8.** A comparison of the performance metrics of the three different wind power plants optimized for different objective functions. These results are shown for the small wind plant with a unidirectional wind rose.

While we included three specific objective functions in this paper, there are many other considerations that could be included in the objective. For example, it may be desirable to maximize the profit generated by each turbine in the wind power plant

above some minimum value. This would keep the optimizer from adding a turbine that only provided a marginal return on investment. One could also optimize for profit or COE while constraining the AEP to be above some desired minimum value. When using mathematical optimization, the objective function must be designed to truly represent the desired performance because this will drive the final solution.

### 6.2.2 Small Power Plant with Unidirectional Wind Rose: Different Algorithms

In this section, we discuss the performance of the solutions found with each optimization strategy and their computational expense. This information is presented in the last five columns of Table 2. As explained previously, the fifth column shows the optimized solution, and the sixth shows the normalized value, to easily see how the optimized solutions compare to each other. The last three columns are measures of the computational expense. The time columns are straightforward and show the total wall time required to run the optimizations. For this paper, everything was run without parallelization on a laptop with a 2.4 GHz 8-core Intel processor. However, just the time as a measure of computational expense may be misleading. There is other overhead in the optimization time other than just objective function calls; therefore, we included a column for total objective function calls and run time, which together give a decent representation of the total computational expense of each algorithm.

For this small wind power plant with the unidirectional wind rose, the greedy algorithm did not perform very well. If found the worst solution for both the AEP and COE objectives, and only the third best solution for the profit objective, but still underperformed by more than 8% compared to the best solution in this objective. This algorithm relies on placing turbines far apart to get the maximum benefit possible at each step of the optimization. Because the domain for this scenario was small, this made it difficult to add additional turbines without violating the spacing constraint. Doing so would require adjusting the location of multiple turbines at once to make room, which is not something this algorithm does. Although the computational expense for the greedy algorithm in this scenario was minimal, its poor performance does not justify its use.

For the objectives with higher turbine density, AEP and profit, the repeated sweep algorithm performed poorly. The answers for these objectives were either the worst or second worst solution found. However, for the COE objective this algorithm performed quite well and found the second best solution, within 0.2% of the best solution. Like the greedy algorithm, the repeated sweep algorithm has a step that relies on greedily placing turbines in the domain if they result in an improvement of the objective function. This algorithm has difficulty placing the turbines without violating spacing constraints for objectives that have many turbines in the optimal solution. For the COE objective however, the optimal number of turbines was much fewer. Thus the repeated sweep algorithm could place the turbines and move them around to a certain extent to find an excellent solution for this objective. The negligible computational expense for this algorithm could justify its use for this scenario, for an objective with optimal turbine spacings that are sufficiently larger than the minimum spacing constraints.

The genetic algorithm with the gridded plant domain performed very well for each objective function. It even outperformed the gradient-based optimization for the COE objective and performed within 3% of the best solution found for the AEP and profit objectives. However, the computational expense was high, requiring the most time of any of the gradient-free algorithms and by far the most function calls. However, because this problem was relatively small, the computational expense was not prohibitive.

The boundary-grid optimization solved with a genetic algorithm performed in the middle of the pack for the gradient-free algorithms. It performed very well with the AEP objective, but poorly with the COE and profit objectives. This was because of the small wind plant area. The boundary-grid formulation forces turbines to be equally spaced around the boundary. With a unidirectional wind rose, this means that some turbines will always be in the back of the power plant relative to the incoming wind. As discussed before, waked turbines were very detrimental for COE and, by extension, detrimental from a profit perspective as well.

Finally, the gradient-based optimizer, while sweeping across the number of wind turbines, performed well for each objective function. Because this algorithm uses continuous design variables and allows full access to the wind plant domain, permitting each turbine to be placed wherever the optimizer deems best, we always expected the gradient-based optimizations to perform well. In fact, we expected this optimizer to perform the best of all for each objective, making it quite surprising that it was outperformed for both the AEP and COE objectives. Even though the gradient-free algorithms do not provide as much freedom as the gradient-based optimization, they were able to find the best solution for these objectives.

At this point, we would like to reiterate that the results shown in Table 2 are for a limited number of starting optimizations. They do not indicate that the solutions found are the best solution that each optimizer is capable of finding. These results simply show the optimizer performance with a small sample size. That said, for a small number of discretizations, finding the optimal number and layout of wind turbines for various objectives can be achieved with a simple genetic algorithm.

## 6.3 Large Power Plant with Unidirectional Wind Rose

The performance and required computational expense of optimization algorithms can vary dramatically depending on the problem size. In this section, we will present another set of optimizations we ran for a larger domain. This wind power plant is a square with 1.6-km sides. For the gridded domain, the plant was divided into a 20-by-20 square grid which, because of the increased size of the plant area, maintained the same spacing between grid points as in the small wind plant example. For a wind plant of this size, we did not run the gradient-based optimizer because of the large computational expense required to run the sweep across all of the possible optimal number of turbines. We only ran the optimizations for the four gradient-free methods we previously discussed. As was done in Sec. 6.2, we ran each optimization method 5 times to convergence for each objective. The wind resource and PPA for this wind power plant were also assumed to be the same as for the small wind plant. The results for this wind power plant optimization are presented similarly to those for the small wind plant, with full results shown in Table 3, and the best layout for each objective shown in Fig. 9.

### 6.3.1 Large Power Plant with Unidirectional Wind Rose: Different Objectives

The general trends that we observed from the smaller wind power plant optimizations hold true for this larger wind plant as well. The wind plant optimized for AEP has as many wind turbines in the plant as possible without violating the spacing constraints, leading to a large number of turbines. The wind plant optimized for COE has turbines that are minimally waked, leading to a very small number of turbines. The wind plant optimized for profit is a middle ground between the previous two objectives. One main difference between the results for this large wind plant and the small wind plant is in the optimal solution

**Table 3.** Complete optimization results for the large wind power plant with unidirectional wind rose.

| objective | optimization | num. turbines | avg. spacing (D) | optimal value | normalized | time (s) | time (hr) | function calls |
|---|---|---|---|---|---|---|---|---|
| AEP (GWh) | greedy grid | 40 | 2.55 | 663 | 0.920 | 746 | 0.21 | 34,012 |
| | genetic grid | 23 | 3.58 | **450 (worst)** | 0.624 | 2,539 | 0.71 | 44,095 |
| | sweep grid | 47 | 2.32 | 710 | 0.985 | 113 | 0.03 | 1,220 |
| | genetic BG | 48 | 2.29 | **721 (best)** | 1.000 | 3,310 | 0.92 | 48,824 |
| COE ($/MWh) | greedy grid | 22 | 3.68 | 20.51 | 1.007 | 423 | 0.12 | 28,393 |
| | genetic grid | 16 | 4.53 | **21.00 (worst)** | 1.031 | 2,656 | 0.74 | 63,104 |
| | sweep grid | 20 | 3.91 | **20.37 (best)** | 1.000 | 119 | 0.03 | 4,895 |
| | genetic BG | 18 | 4.19 | 20.73 | 1.018 | 1,002 | 0.28 | 35,814 |
| annual profit ($MM) | greedy grid | 31 | 2.97 | 4.98 | 0.979 | 592 | 0.16 | 31,902 |
| | genetic grid | 20 | 3.91 | **3.33 (worst)** | 0.655 | 2,826 | 0.79 | 47,977 |
| | sweep grid | 31 | 2.97 | 4.93 | 0.970 | 116 | 0.03 | 2,528 |
| | genetic BG | 33 | 2.86 | **5.09 (best)** | 1.000 | 2,126 | 0.59 | 52,869 |

for maximum profit. For the large plant, the wake losses for the optimal profit solution are significantly higher than for the small wind plant, 12% compared to 5.4% (see Figs. 10 and 8). More turbines can fit within the boundary of the large plant, pushing the plant further down the BOS cost curve shown in Fig. 3. The reduced costs from economies of scale make up for higher wake losses in the optimal solution.

Figure 10 shows several metrics for the three wind power plants that were optimal for the different objective functions. Although the trends in this figure are similar to those in Fig. 8, the differences in metrics are more extreme for the different objective functions. While all of the metrics are interesting, there are a few specific observations that are worth pointing out. First, the COE for the wind plant optimized for profit is relatively low, which is impressive because COE was never directly minimized. Second, the profit for the wind plant optimized for AEP is very low. Even though the plant produces a lot of energy, the tight turbine spacing causes high wake losses and inefficient turbines; thus, the high costs are not offset by the high energy production. Third, the wake losses for the wind plant optimized for minimum COE are very low, less than 2%. This is impressively low and is only possible because of the unidirectional wind rose.

## 6.3.2   Large Power Plant with Unidirectional Wind Rose: Different Algorithms

While the general trends found for the solutions with different objectives were similar between the small and large wind power plants, the computational expense and performance of different algorithms were not. In the last five columns of Table 3, we see how well each optimization method performs. The most glaring difference is seen in the performance of the genetic algorithm with the gridded turbine domain. While with the small wind plant, this method provided the best or near-best results for each

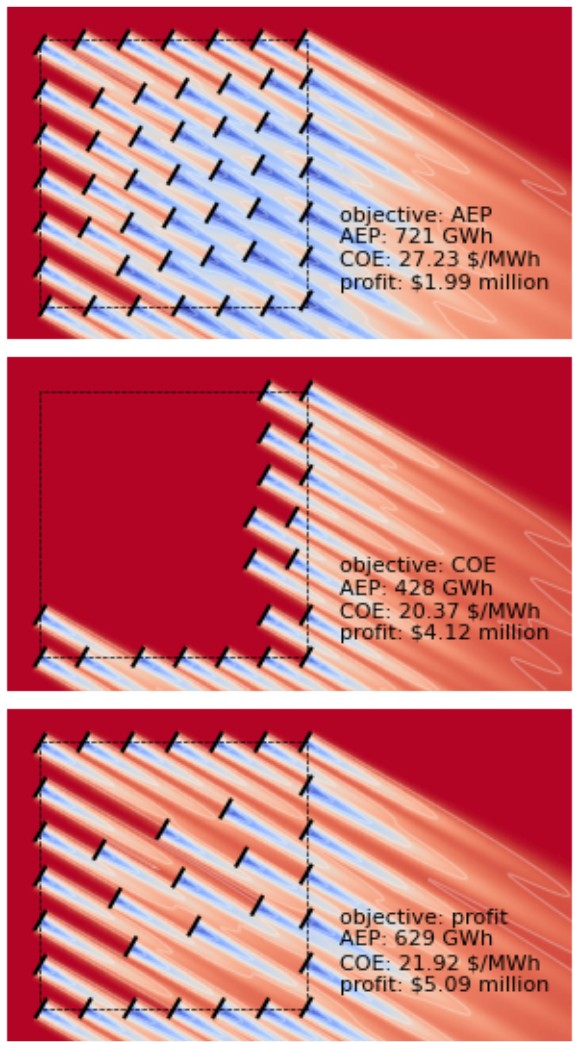

**Figure 9.** The optimal layouts for each objective for the large wind power plant with a unidirectional wind rose. From top to bottom, the associated objective functions are: AEP, COE, and profit. The text within each figure provides the value of all three metrics for each wind plant.

objective, in this larger domain it severely underperformed compared to the other optimization methods. While the genetic algorithm was easily able to handle the 100 design variables from the 10-by-10 grid of the small wind plant, it appears unsuited to find a good solution for the 20-by-20 grid of the large wind plant. This observation agrees with our previous intuition with genetic algorithms in that they tend to perform poorly as problem complexity increases.

Next, the greedy algorithm performs relatively better in the larger wind plant than it did in the smaller wind plant. For the AEP objective, this algorithm still underperformed compared to the best solution, for the same reason as in the small plant.

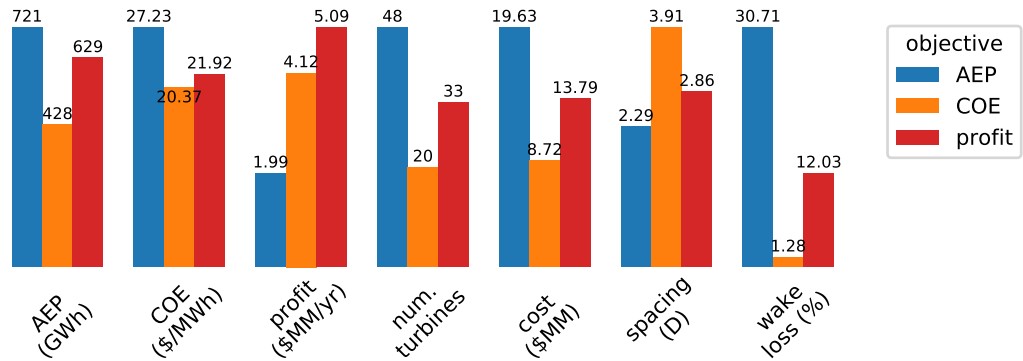

**Figure 10.** A comparison of the performance metrics of the three different wind power plants optimized for different objective functions. These results are shown for the large wind plant with a unidirectional wind rose.

However, for the other objectives the greedy algorithm found a very good solution. Within 1% of the best solution for the COE objective, and within 2.1% of the best solution for the profit objective. Like the greedy algorithm, the repeated sweep algorithm performed relatively better in the larger wind plant than it did in the smaller wind plant. In fact, this algorithm performed well for each objective, finding a solution within 1.5% of the best for the AEP objective, the best solution for the COE objective, and within 3% of the best for the profit objective.

In addition to the optimal solution, it is also important to analyze the computational expense for the greedy and repeated-sweep algorithms. In the gridded formulation, the number of design variables quadruples when going from the 10-by-10 grid to the 20-by-20 grid. However, the computational expense increases much more than a factor of four between these two scenarios. For the greedy algorithm, the required time increased roughly 40–50 times between the small and large scenarios, and the required function calls increased about 15 times. For the repeated sweep algorithm, the required time increased by a factor between 15–30, while the required function calls increased by a factor of 6–7.5. This is an indication of the poor scaling of computational expense with increasing design variables for gradient-free algorithms. It is a combination of each iteration requiring more function calls, and each function call requiring more time with the larger domain.

Finally, the boundary-grid algorithm performed excellently for each objective function for this large wind power plant optimization. The organized structure forced by the boundary-grid method was able to fully take advantage of the unidirectional wind rose and create layouts with the appropriate number of turbines where waking is minimal. This method found the best solution for the AEP and profit objectives, and was within 2% of the best solution for the COE objective. Even though it used a genetic algorithm, the boundary-grid optimization performance did not suffer with the increased size of the wind plant. With this formulation, the number of design variables remains constant, independent of the number of turbines being modelled and the size of the domain. There was still an increase in the computation time between the small and large domain scenarios, but this was because each function call requires more time. The total number of function calls between the scenarios remained fairly constant.

## 6.4 Large Power Plant with Full Wind Rose

The final scenario in which we examine the performance of each optimization algorithm is for the large wind power plant domain. Unlike the previous examples, the optimization results shown in this section include a full wind rose, discretized into 72 wind direction bins. As shown in Fig. 11, the most probable wind directions are from the west and south-southeast. The wind rose we used for this scenario is similar to the one used in the first International Energy Agency (IEA) Task 37 wind plant layout optimization case study, with more finely discretized bins (Baker et al., 2019). The wind rose was specifically chosen to have a dominant wind direction in line with the upper and lower wind plant boundaries. Because the boundary-grid method is formulated to place turbines around the wind plant boundary, we wanted to see how it would perform with an unfavorable wind rose. Everything else for this optimization scenario was the same as in Sec. 6.3, including the wind plant dimensions, the grid discretizations, number of random starts, and PPA. The results for this section are shown in Table 4 and Fig. 12. Because the wind resource was unidirectional in the previous wind power plant layout figures, only one figure was needed to represent the flow field associated with each layout. Because this section includes optimization results that were run with a full wind rose, we show two wind directions in Fig. 12, with each column representing the two dominant wind directions of west and south-southeast.

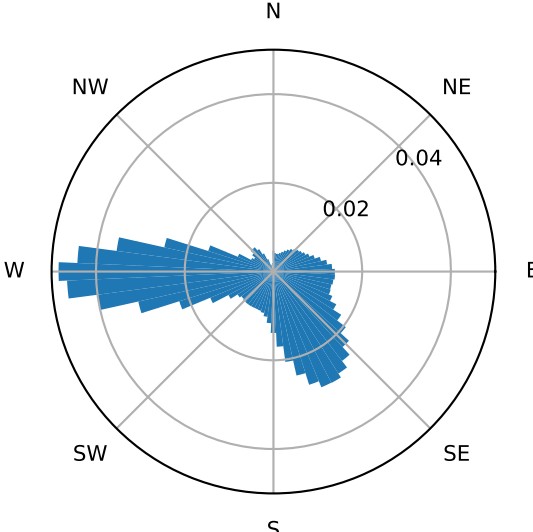

**Figure 11.** The full wind rose used for the optimizations in this study. The wind rose is divided into 72 bins with dominant wind directions from due west and from south-southeast.

**Table 4.** Complete optimization results for the large wind power plant with full wind rose.

| objective | optimization | num. turbines | avg. spacing (D) | optimal value | normalized | time (s) | time (hr) | function calls |
|---|---|---|---|---|---|---|---|---|
| AEP (GWh) | greedy grid | 43 | 2.44 | 582 | 0.928 | 38,329 | 10.65 | 35,680 |
| | genetic grid | 26 | 3.31 | **429 (worst)** | 0.685 | 37,208 | 10.34 | 48,739 |
| | sweep grid | 50 | 2.24 | 604 | 0.964 | 5,560 | 1.54 | 1,166 |
| | genetic BG | 54 | 2.14 | **627 (best)** | 1.000 | 157,191 | 43.66 | 41,175 |
| COE ($/MWh) | greedy grid | 14 | 4.95 | 22.61 | 1.002 | 10,434 | 2.90 | 21,561 |
| | genetic grid | 11 | 5.86 | **22.90 (worst)** | 1.015 | 28,386 | 7.89 | 46,823 |
| | sweep grid | 15 | 4.73 | **22.57 (best)** | 1.000 | 7,400 | 2.06 | 7,812 |
| | genetic BG | 12 | 5.51 | 22.74 | 1.007 | 25,260 | 7.02 | 37,071 |
| annual profit ($MM) | greedy grid | 23 | 3.58 | 2.68 | 0.967 | 22,020 | 6.12 | 29,538 |
| | genetic grid | 19 | 4.04 | **2.27 (worst)** | 0.821 | 34,824 | 9.67 | 47,266 |
| | sweep grid | 24 | 3.48 | **2.77 (best)** | 1.000 | 8,277 | 2.30 | 4,201 |
| | genetic BG | 24 | 3.48 | 2.59 | 0.936 | 60,995 | 16.94 | 40,959 |

### 6.4.1 Large Power Plant with Full Wind Rose: Different Objectives

Again, many of the general trends for the different objectives are the same as the results for the unidirectional wind rose cases. In this section, we only focus on the key differences between the results for the large wind power plant with a unidirectional wind rose and the results for the large wind plant with the full wind rose. There are two observations we would like to point out.

First, and most importantly, the optimal number of turbines for the COE and profit objectives is much lower for the full wind rose than for the unidirectional rose. For the unidirectional rose, the optimal numbers of turbines for COE and profit are 20 and 33, respectively. For the full wind rose, these numbers are 15 and 24, respectively. With the full wind rose, turbines will always be unfavorably waked for some wind directions. It is impossible to find a turbine layout in which all of the turbines are always unwaked. For a unidirectional rose, the turbines can be very closely spaced in directions not aligned with the wind and still avoid wakes and perform well in each objective. This luxury does not exist for a full wind rose. Turbines spaced close together will perform poorly at least some of the time. Therefore, fewer turbines are placed in the optimal wind plants with a full wind rose so that the average spacing can be higher and reduce the wake effects.

Second, notice that the optimal number of turbines for the AEP objective is higher for the full wind rose (54 turbines) than for the unidirectional wind rose (48 turbines). This is simply an artifact of our methodology. For the AEP objective, the unidirectional wind rose would also benefit from having more turbines spaced closer together. However, the optimizer didn't find this solution from the five optimizations that we ran. Setting up an optimization run always involves a trade-off between trying to find the best solution and minimizing computational expense. One can imagine two extremes for a genetic algorithm.

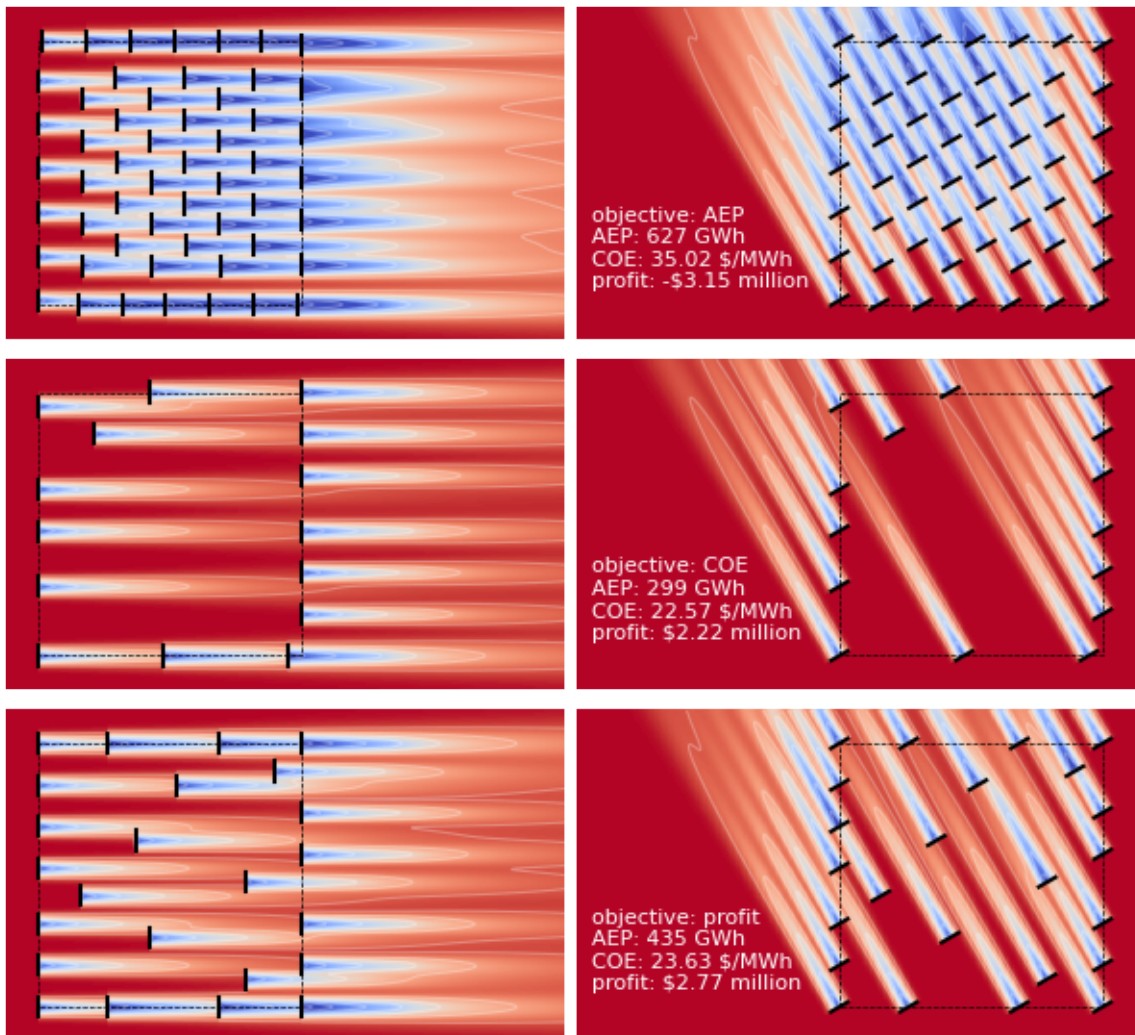

**Figure 12.** The optimal layouts for each objective for the large wind power plant with a full wind rose. From top to bottom, the associated objective functions are AEP, COE, and profit. The text within each figure provides the value of all three metrics for each wind plant. The two columns represent the flow field for each objective from the two dominant wind directions.

The first extreme has an enormous population size and very strict convergence criteria. This optimization would theoretically find a very good, maybe the best solution, but at a restrictively high computational expense. The other extreme would have a minuscule population and very lax convergence criteria. This population would converge very quickly, but would lend very little confidence that a good solution was found. For this paper, our goal was to examine overall trends, and not to find the global solution for every scenario and optimizer combination. For a one-off optimization, it may be prudent to run more than

5 optimizations with different initialization of the design space, and maybe tune the optimizer parameters and convergence

criteria to the specific problem. However, for this paper our goal was to run a large quantity of optimizations across a range of scenarios, which required us to make some decisions to keep the computational expense reasonable. Even though the optimizers for the large wind farm and unidirectional wind rose scenario failed to find the best solution for the AEP objective, we can stand behind our methodology and have confidence that the general trends we have observed are accurate and valid.

Similar to Fig. 8 and Fig. 10, Fig. 13 shows the different wind power plant performance metrics for each optimal wind plant with the different objectives. The general trends are similar to Fig. 10 with a few differences. First, and most apparent, the profit for the wind plant optimized for AEP is extremely low. In fact, this wind plant loses more than $3 million each year because both the costs and wake losses are so high. There are at least two reasons for this extreme negative value for profit: 1) Our cost model has low economies of scale. At large numbers of turbines, the cost reductions for adding more turbines is low. And, 2) our minimum spacing constraints for this paper were small, only two rotor diameters. At this close spacing, the wake losses are high. For wind plants where the minimum spacing is much larger (5 or more rotor diameters), a wind plant optimized for AEP may still perform well in the other objectives. The second difference in general trends is that the wake losses for these wind plants are much higher than for the optimal wind plants and the unidirectional wind rose. The reason for this was previously mentioned—with multiple wind directions, there will always be some wind directions for which a turbine operates within a wake.

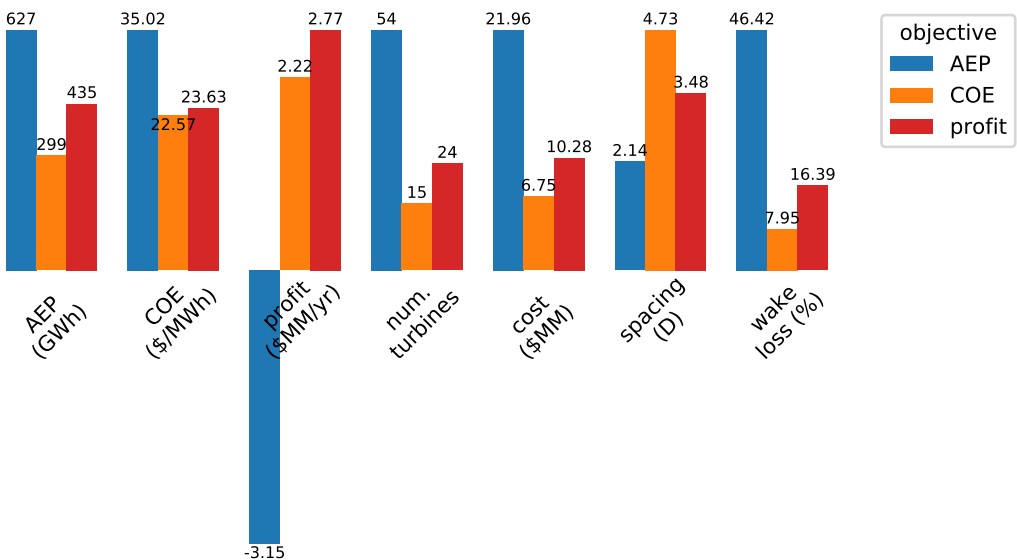

**Figure 13.** A comparison of the performance metrics of the three different wind power plants optimized for different objective functions. These results are shown for the large wind plant with a full wind rose.

### 6.4.2 Large Power Plant with Full Wind Rose: Different Algorithms

There are three main differences we observe between the unidirectional wind rose algorithm performances and those for the full wind rose. First is the expected increase in computational time. The function calls for each algorithm and for each objective are very close for the different wind roses. However, the computational time for the full wind rose is significantly and expectedly higher. The computational expense of each function call for the full wind rose is about 72 times that of the unidirectional rose—72 being the number of wind direction bins. The reason the time does not scale exactly linearly with the number of function calls is the certain amount of infeasible layouts that are produced during the optimization that do not call the full wake model. These calls are very fast, but not instantaneous, and do add up over time.

The second observation is about the impressive performance of the boundary-grid problem formulation, even though the wind rose is specifically selected to put this method at a disadvantage. As expected, the boundary-grid method tremendously outperformed the other methods for the AEP objective because this formulation can add many turbines to the wind power plant with very small adjustments to the design variables. For the COE objective, the boundary-grid optimization performed well, within a percent of the best solution found. The boundary-grid optimization performed the worst for the profit objective, about 6.5% worse than the best solution found. For many wind plants, the boundary-grid method would perform much better because prominent wind directions will not always be aligned with the wind plant boundaries.

The third and final observation is about the performance of the repeated-sweep algorithm. For the AEP objective, this algorithm performed worse than the boundary-grid method, but still produceed a wind power plant layout that performed favorably compared to the overall optimal, within 4%. The repeated-sweep algorithm is ineffective at placing as many turbines as possible into the wind plant, but still performed relatively well. For the COE and profit objectives, the repeated-sweep algorithm found the best solution. For the COE objective, the greedy, repeated-sweep, and boundary-grid optimizations all performed similarly. However, for the profit objective, the repeated-sweep algorithm impressively outperformed all of the other algorithms. With a full wind rose and objectives that favor solutions where the turbines are minimally waked, this algorithm performed extremely well. Because the best solutions have turbines that are spaced farther apart, the optimizer is able to search the design space without violating the turbine spacing constraints. In addition to finding superior optimal solutions, the repeated sweep algorithm required much less computational expense. For this problem size, the repeated-sweep algorithm performed the best overall. In short, the boundary-grid method performed relatively well even with an unfavorable wind rose and boundary. The repeated-sweep algorithm performed well for the COE and profit objectives and optimized very quickly compared to the other methods.

### 6.5 Overall Algorithm Performance

In this section, we discuss the overall performance of each optimization algorithm for the small wind power plant with a unidirectional wind rose, the large wind plant with the unidirectional wind rose, and the large wind plant with the full wind rose. Figure 14 shows the overall performance of each algorithm and the computational expense for each objective and size of

wind plant. Much of this information has been discussed in previous sections, but we review each of the algorithms here from an overall perspective.

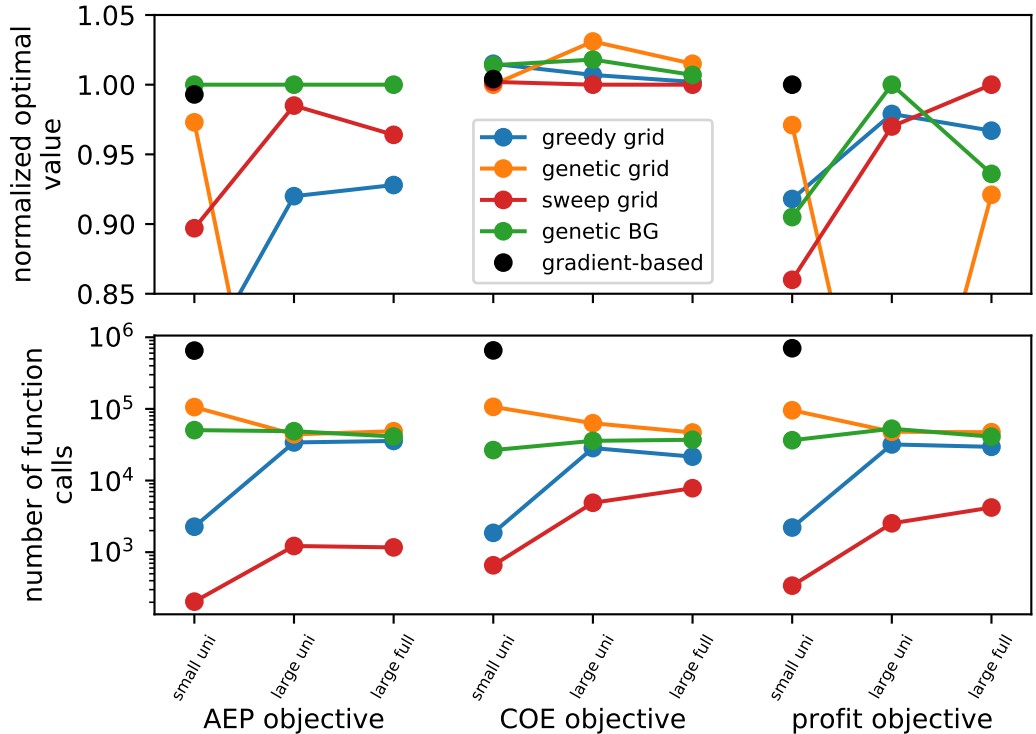

**Figure 14.** Overall performance representations of the different optimization algorithms. The different colors represent the different optimization methods. The top figure shows the best solution quality, and the bottom figure shows the number of function calls required.

650    The greedy algorithm performed poorly for the small wind power plant and better for the larger wind plant, although it never performed the best. The computational expense for this algorithm was very low for the small wind plant, but greatly increased for the large wind plant. We can conclude that the computational expense of the greedy algorithm scales poorly with increasing design variables.

    The genetic algorithm with the gridded plant domain performed very well for the small wind plant and was close to the best

655  solution for each objective. However, for the larger wind plants, this optimization method performed very poorly. We want to be reiterate that part of the reason this optimization method performed so poorly was because we kept the convergence criteria the same for the small and large wind plants. Individual parameters could be altered to get better performance with the gridded domain and the genetic algorithm; however, it is unlikely that it would perform as well as the other algorithms. Genetic algorithms typically have a hard time finding a good solution with large numbers of design variables.

660    Like the greedy algorithm, the repeated-sweep algorithm performed comparatively poorly for the small wind power plant, but much better for the large wind plants. In fact, this algorithm achieved the best, or close to the best, solution for all of the objectives for each of the large wind plant optimizations. In addition, the computational expense for this algorithm was by far the lowest for the problem sizes included in this paper. That said, similar to the greedy algorithm, the number of function calls required for the repeated-sweep algorithm increased greatly from the small wind plant to the large wind plants. Even though

665    it was computationally efficient for the problem sizes in this paper, as the problem grows, the computational expenses would grow as well and eventually limit the size of wind plant that can be optimized.

    The boundary-grid optimization consistently performed the best for the AEP objective. It generally performed poorly for the COE objective, and its performance for the profit objective varied. For the large wind power plant and unidirectional wind rose, the boundary-grid method found the best solution for the profit objective. However, with the full wind rose, the boundary-

670    grid method performed the worst for the profit objective. This is largely because, for our specific case, the full wind rose had a dominant wind direction directly in line with the wind plant boundaries. This will not always be the case; for most other scenarios, the boundary-grid method would perform comparatively well. A primary reason that the boundary-grid method is so effective is that the required function calls do not change with the problem size. While the greedy and repeated-sweep algorithms required many more function calls as the wind plant domain increases in size, the boundary-grid method remained

675    constant between the small and large wind plants. At some point, the boundary-grid method would be more computationally efficient than any of the other algorithms because the number of design variables always remains constant.

### 6.6    Varied Power Purchase Agreement

The final set of results we present in this paper explores how the optimal profit solution is affected by the PPA. For this section, the problem formulation was identical to Sec. 6.4, including using the full wind rose shown in Fig. 11, the wind speed (10

680    m/s), the wind power plant size (1600-by-1600 $m^2$), grid discretization (20-by-20), and the number of randomly initialized optimizations run (5). However, in this section, instead of assuming a PPA of \$30/MWh, we repeated the optimization while varying the PPA from \$25/MWh–\$100/MWh. Because the repeated-sweep algorithm performed the best for the profit objective in Sec. 6.4, combined with this algorithm's low computational expense, we performed these optimizations only with the repeated-sweep algorithm and did not compare with the performance of the other algorithms. The results for these optimizations

are shown in Fig. 15. From top to bottom, this figure shows the optimal number of turbines, profit (which was the objective of the optimizations), AEP, and COE as a function of the PPA.

    As the value of the energy produced increased (represented by increasing the PPA), the optimal number of turbines in the wind plant also increased. With a low PPA, the optimal solution resembles the optimal COE solution from Sec. 6.4. Wakes are avoided as much as possible, and the number of turbines in the wind plant is low. With a higher PPA, the solution approaches

the optimal AEP solution. Gains in AEP can be worthwhile with higher PPA—even if they come at the expense of reducing the overall efficiency of each individual turbine.

    Some typical PPAs in the United States are shown in Fig. 16, which includes the levelized PPA for various projects in the United States since 2010. For the data shown in this figure, the levelized PPA does not take into account any federal tax credit.

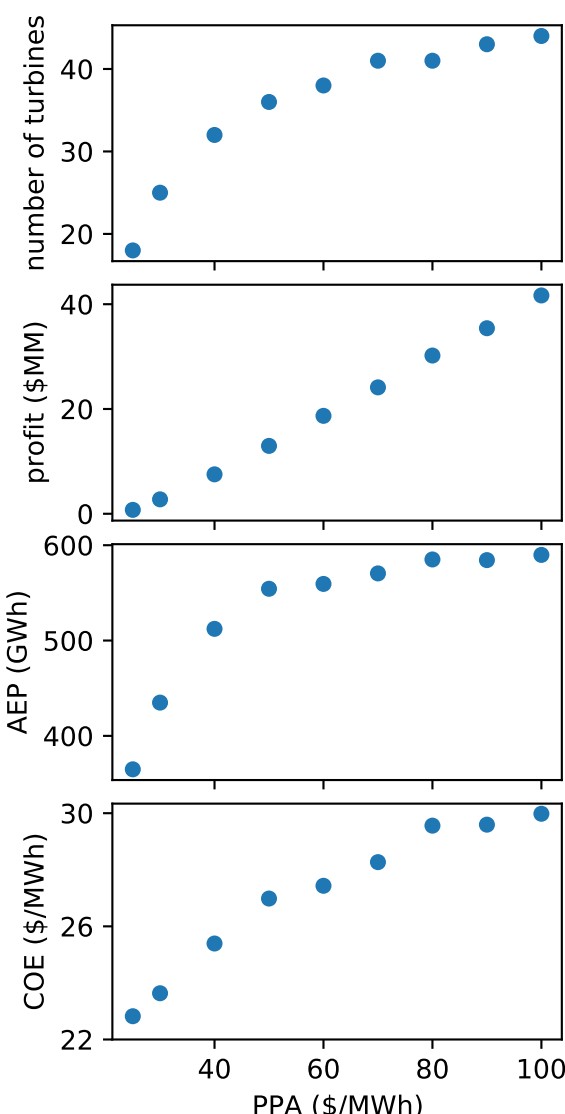

**Figure 15.** The effect of PPA on the optimal solution for wind power plants optimized for profit. From top to bottom are the number of turbines, profit, AEP, and COE as a function of the PPA.

The different colors represent projects in different parts of the country, which indicates that projects in the central states tend to have lower PPAs. A decade ago, the range of PPAs in the United States was quite large, from around $40/MWh all the way up to $120/MWh; however, more recent PPAs are much lower, closer to $20/MWh–$40/MWh. This decrease in PPA prices in recent years has been the combined result of higher capacity factors, declining installation costs and operating costs, and low interest rates (Wiser and Bolinger, 2019).

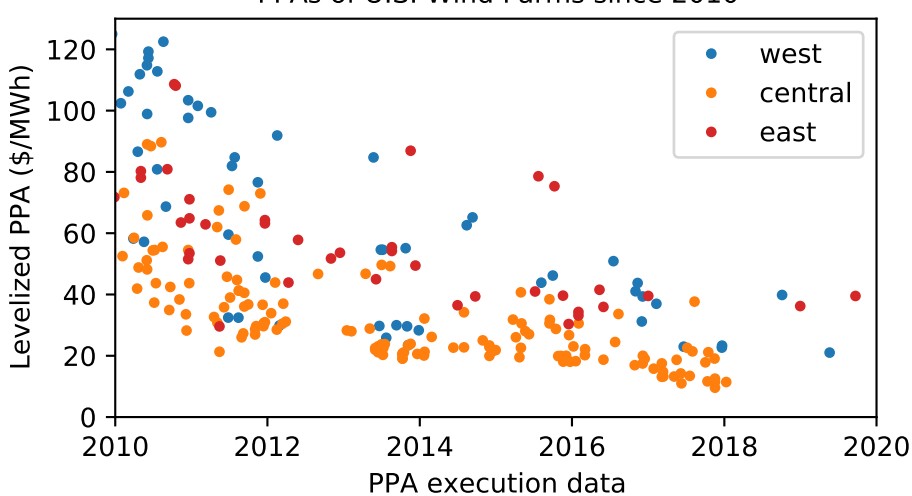

**Figure 16.** The PPAs for wind power plants built in the United States since 2010. Different colors represent wind plants in different regions of the United States.

## 7 Further Discussion

### 7.1 Empirical Considerations

The wind power plant layout optimizations in this paper assumed a fixed land area in which the wind turbines must remain inside. This is typically referred to as a land-constrained site and empirically represents a scenario where terrain, resource availability, social and siting considerations, or other factors limit the amount of land that is available for wind turbine installations. Although there are many sites in the United States that are capacity constrained (limited by the capacity of the transmission interconnection), the wind plants optimized for this paper assume land-constrained sites, which are also found throughout the United States. Examples of land-constrained sites in the United States with high turbine density are New England and San Gorgonio, California. These high turbine densities are driven by high PPAs in New England and exceptional wind resources and high-capacity transmission in San Gorgonio.

Land availability, combined with installed project cost, project size, power purchase prices, and others, all influence capacity density. Another driving factor that can determine the capacity density of a given wind plant is market competition, which was not considered in this analysis. Wind plant developers and owners have target profitability values that they wish to achieve. In any energy market, the competition between developers will result in a range of profitability values that results in limited variation in capacity for a specific location. Variations in PPA prices for actual projects are typically due to variations in resource, cost, tax policy, and investment strategies.

As shown in Fig. 15, as PPA price increases for a constant project area, so does the optimal number of turbines. At high turbine densities, the AEP gain from adding additional turbines is minimal because of large wake losses. The minimal gains of AEP can justify the extra costs in these scenarios because the value of the small amount of additional electricity is so high. The optimal solutions that we found for our results greatly depend on the assumptions and models we used.

## 7.2    Overview of Optimization Cases

In Sec. 6, we present a one-dimensional wind farm sweep, a small wind farm layout optimization with a unidirectional wind rose, a large wind farm layout optimization with a unidirectional wind rose, and a large wind farm layout optimization with a full wind rose. For each of these scenarios, we assumed a constant wind speed, a square wind farm domain, a constant PPA, a relatively small minimum spacing constraint of 2 rotor diameters, a constant turbine capital cost, and a one-dimensional relation between BOS costs and wind farm capacity. These modeling assumptions were made to reduce the complexity of the problem, and identify key relationships and trends between different scenarios. Future work could benefit from increasing the fidelity and realism of these assumptions, such as including full wind direction and wind speed distributions in the wind resource, exploring more realistic wind farm boundaries, and testing the sensitivity of the results to the minimum spacing constraint. Also it is important to note that the work presented in this paper is for land constrained sites. These results could change dramatically for sites without a fixed wind farm boundary.

The optimization parameters for each optimizer were set with a combination of trial and error, and best practice recommendations. As with any optimizer, with the exception of brute force testing every possible combination of design variables, there was some randomness in the solutions that were found. This randomness comes from the starting point for the greedy algorithm, the starting point and order of sweeping through the space for the repeated sweep algorithm, and the starting population, the crossover points, and the mutations for the genetic algorithms. As with many optimization algorithms, this randomness is inherent in the optimizer. To reduce the variability and converge on a solution close to the global optimum, several runs should be made with different initializations of the design variables. For this study, we repeated each optimization five times. Although this is much better than a single optimization run, better results may be obtained by performing more random starts. In future research, it may be beneficial to consider fewer scenarios, but perform more random starts.

## 8    Conclusions

In this paper, we present our work on wind power plant layout optimization, which includes optimizing the number of wind turbines. We specifically discuss the effect of different objective functions on the optimal solution, as well as the pros and cons associated with using different problem formulations and optimizers to solve the problem. We explore optimizing several different wind plants for objectives of AEP, COE, and profit. We found that the number of turbines in each optimal solution was highly sensitive to the objective function that was used. The plants that we optimized for AEP tended to have the most wind turbines, while those optimized for COE had the least, and those optimized for profit were somewhere in between. The purpose of this paper is not to provide an optimal wind plant layout for a specific wind plant boundary and wind conditions,

nor is it to suggest general rules of thumb for designing wind plants with different objective functions. The purpose is to clearly demonstrate that the solution from optimizing a wind plant can be heavily influenced by the objective function, particularly when considering the number of wind turbines as a design variable. Specific solutions and layouts should be determined by the models and problem parameters, and the objective must be carefully chosen to represent the desired outcome because any mathematical optimizer will exploit this objective. From the models we used in this paper, the optimal number of turbines for a square wind plant with a full wind rose was 15, 24, or 54, depending on the objective of COE, profit, or AEP, respectively.

The other area that we discuss in this paper is problem formulation and algorithm selection for performing the optimization of turbine number and wind power plant layout optimization. We also present a very simple repeated-sweep algorithm that performs well, especially for the larger design spaces and for a full wind rose. For a coarse wind plant discretization, in this paper we used a 10-by-10 grid and found that a simple genetic algorithm performed extremely well in selecting turbine number and location, even compared to a gradient-based optimizer. However, for a 20-by-20 grid, the genetic algorithm performed poorly. A greedy algorithm and the presented repeated-sweep algorithm performed well for the COE and profit objectives, particularly with a full wind rose leading to turbines that are spaced farther apart on average. At least for the wind plant sizes that we used in this paper, the computational expense of the greedy and repeated-sweep algorithms was comparatively low, even for cases with more grid points. The boundary-grid problem formulation optimized with a simple genetic algorithm performed well regardless of the size of the wind plant, but it performed comparatively better for the larger wind plant size. Another benefit of the boundary-grid method is that the number of function calls required to optimize the plant stayed relatively constant as the size of the wind plant domain changed. As long as the time required for the function calls is reasonable, or can be made reasonable, optimizing with the boundary-grid method will produce a layout that performs well regardless of the objective.

*Code availability.* All of the code that was used for this paper is publicly available. The repository for this paper, including the optimizers, run scripts, objective functions, and figure generation can be found at: https://github.com/pjstanle/stanley2020-turbine-number. The FLORIS wake model can be found at: https://github.com/NREL/floris.

*Author contributions.* APJS led this research, including developing and testing optimizers, running the optimizations, generating figures, and writing the paper. OR helped develop ideas, provided data for the cost models, and wrote and provided feedback on portions of the paper. JK and CB provided ideas and guidance for the research as well as editing and feedback on the writing.

*Competing interests.* The authors declare no competing interests.

*Acknowledgements.* This work was authored by the National Renewable Energy Laboratory, operated by Alliance for Sustainable Energy,
LLC, for the U.S. Department of Energy (DOE) under Contract No. DE-AC36-08GO28308. Funding provided by the U.S. Department
of Energy Office of Energy Efficiency and Renewable Energy Wind Energy Technologies Office. The views expressed in the article do
not necessarily represent the views of the DOE or the U.S. Government. The U.S. Government retains and the publisher, by accepting the
article for publication, acknowledges that the U.S. Government retains a nonexclusive, paid-up, irrevocable, worldwide license to publish or
reproduce the published form of this work, or allow others to do so, for U.S. Government purposes.

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
