# Peer review of "Objective and Algorithm Considerations When Optimizing the Number and Placement of Turbines in a Wind Power Plant"

_Wind Energy Science, 2021_

## Referee Comment (RC2)

[revised manuscript text omitted]
. As far as we are aware, this is the first paper that starts to explore the effect of the objective on the final wind power plant design. The objective can greatly affect the final layout and performance of an optimized wind power plant. Second, we compared using different problem formulations and optimization algorithms in finding a solution. Previous papers have begun

70 to research this topic but, in this paper, we specifically study how different algorithms perform depending on the objective and the size of the optimization problem. We compare a genetic algorithm and a greedy algorithm in a gridded wind power plant domain, two commonly used wind power plant optimization methods, as well as a genetic algorithm with the boundary-grid method and a new repeated-sweep algorithm in a gridded domain.

**2   Wake Model**

75 The wind speed downstream of a turbine is reduced because turbines extract energy from the flow and from the complex physics of the wakes they produce. In this paper, the desirability of the wind power plants we examined were dependent, to a large extent, on energy production. This energy production is a function of the wind speeds throughout the wind power plant. To calculate the wind speeds to be used in turbine power calculations, we used an analytic Gaussian wake model (Bastankhah and Porté-Agel, 2016; Abkar and Porté-Agel, 2015; Niayifar and Porté-Agel, 2016). The wake calculations were performed

80 using FLOw Redirection and Induction in Steady State (FLORIS), which is a computationally inexpensive, controls-oriented tool to calculate the steady-state flow field in a wind power plant (NREL, 2020). We include a brief description of the Gaussian wake model in this paper but, for more details, refer to the original model paper (Bastankhah and Porté-Agel, 2016).

    Using the Gaussian wake model, the velocity of the wake behind a turbine is computed with the following analytical expressions:

85
$$\frac{u(x,y,z)}{U_\infty} = 1 - Ce^{-(y-\delta)^2/2\sigma_y^2 - (z-z_h)^2/2\sigma_z^2} \tag{1}$$

$$C = 1 - \sqrt{1 - \frac{(\sigma_{y0}\sigma_{z0})C_T}{\sigma_y\sigma_z}} \tag{2}$$

where $u$ is the velocity at a desired location $(x,y,z)$, $C$ is the velocity deficit at the wake center, $U_\infty$ is the freestream velocity, $\delta$ is the wake deflection, $z_h$ is the hub height of the turbine, $\sigma_y$ defines the wake width in the $y$ direction, and $\sigma_z$ defines the

90 wake width in the $z$ direction. Each of these parameters is defined with respect to each turbine. The subscript "0" refers to the initial values at the start of the far wake, which is dependent on ambient turbulence intensity, $I_0$, and the thrust coefficient, $C_T$.

[Figure]

[Figure]

For additional details on near-wake calculations, refer to the original paper describing this model (Bastankhah and Porté-Agel, 2016). The velocity distributions $\sigma_z$ and $\sigma_y$ are defined as:

$$\frac{\sigma_z}{D} = k_z \frac{(x - x_0)}{D} + \frac{\sigma_{z0}}{D} \quad \text{where} \quad \frac{\sigma_{z0}}{D} = \frac{1}{2}\sqrt{\frac{u_R}{U_\infty + u_0}} \tag{3}$$

95

$$\frac{\sigma_y}{D} = k_y \frac{(x - x_0)}{D} + \frac{\sigma_{y0}}{D} \quad \text{where} \quad \frac{\sigma_{y0}}{D} = \frac{\sigma_{z0}}{D} \cos\gamma \tag{4}$$

where $D$ is the rotor diameter, $u_R$ is the velocity at the rotor, $u_0$ is the velocity behind the rotor, $\
[revised manuscript text omitted]

10:         **if** $\texttt{g} < \texttt{m}$ **then**

11:             redefine the iteration minimum: $\texttt{m} = \texttt{g}$

12:             define the iteration minimum array: $\texttt{C} = \texttt{B}$

13:         **end if**

14:     **end for**

15:     **if** $\texttt{m} = \texttt{f}$ **then**

16:         converged

17:     **else**

18:         redefine the best fitness value: $\texttt{f} = \texttt{m}$

19:         redefine the best set of design variables: $\texttt{A} = \texttt{C}$

20:     **end if**

21: **end while**
* * *
**Output:**

    best fitness value: `f`

    best set of design variables: `A`
* * *
[Figure]
* * *
**Algorithm 2** Genetic Algorithm
* * *
**Input:**

    number of bits: `n`

    objective function: `obj`

    population size: `pop_size`

    maximum generations: `n_gens`

    crossover rate: `crossover`

    mutation rate: `mutation`

    tolerance: `tol`

[revised manuscript text omitted]
 consider the price of power and land availability but assume a constant power density for a single turbine configuration and do not vary with PPA price or cost. This work illustrates not only the difference in capacity density for a COE minimization versus a profit maximization, but also the relationship of increasing energy prices on capacity in land-constrained scenarios. Aggressive carbon reduction scenarios or other renewable energy goals typically result in higher energy prices for renewables. These results in Fig. 5 show that for scenarios where energy prices increase, capacity densities may also increase. Similarly, where project costs increase because of smaller plant sizes and higher construction costs, projects may tolerate lower energy production from lower wind resource quality and higher wake losses in order to maintain a constant PPA price.

**6.2 Small Plant with Unidirectional Wind Rose**

With the 1D sweep of the design space complete to provide some intuition about the different objective functions, we will now discuss a simple layout optimization for a small wind power plant with a unidirectional wind rose. As stated before, we performed the optimization of each objective using a gridded domain, optimized with a greedy algorithm, a genetic algorithm, and a repeated sweep algorithm. We also optimized a boundary-grid layout parameterization with a genetic algorithm. Also, as mentioned before, for this small wind plant, we optimized the layouts using gradient-based optimization.

For this small wind power plant optimization, we assumed the domain was square with 800-m sides. The wind came from a direction of 300 degrees, or 30 degrees north of west. The wind speed was assumed constant at 10 m/s, which is close to the

355 rated wind speed for our turbine model. The PPA was assumed to be $30/MWh. For the gridded design variables, the domain was discretized into a 10-by-10 grid. We ran each optimization method five times to convergence because the final solution is dependent on the randomly initialized population or design space. Because each of the optimization algorithms has some stochastic qualities, with enough time and randomly initialized starts, each optimization method will potentially be able to find a very good solution. However, we believe that five optimizations for each is enough to give a good idea of their performance

360 relative to each other for each of the objective functions.

Results for the small wind power plant optimizations with a unidirectional wind rose are shown in Table 2 and Fig. 6. Table 2 shows the optimization results and the computational expense associated with each optimization method and for each objective function. The first column shows the objective function, and the second column shows the optimization method. The third column provides the optimized number of turbines in the wind plant in bold and the average turbine spacing in rotor diameters

365 associated with that number of turbines. The fourth column provides the best solution from the 5 optimizations in bold, along with this same value normalized by the best solution out of all of the optimization methods for the given objective. In this column, the best solution is highlighted in blue and the worst solution is highlighted in red. Finally, the fifth and sixth columns provide the total time and the total number of function calls to run the 5 optimizations for each optimization method. The gray rows in this figure show the gradient-based optimization results. Notice one cell in a gray row is blue, indicating the gradient-

370 based optimization found the best solution for the maximum profit objective. 
[revised manuscript text omitted]
 three columns of Table 2. As explained previously, the third to last column shows the optimized solution in bold, as well as the normalized value, to easily see how the optimized solutions compare to each other. The last two columns are different measures of the computational expense. The time column is straightforward and is the total wall time required to run the optimizations. For this paper, everything was run without parallelization on a laptop with a 2.4 GHz 8-core Intel processor. However, just the time as a measure of computational expense may be misleading. There is other overhead in the optimization time other than just objective function calls; therefore, we included a column for total objective function calls and run time, which together give a decent representation of the total computational expense of each algorithm.

For this small wind power plant with the unidirectional wind rose, the greedy and repeated sweep algorithms do not perform very well. Although they are by far the best when it comes to required computational resources, both algorithms find comparatively poor solutions. Both of these algorithms rely on placing turbines far apart to get the maximum benefit possible at each step of the optimization. Because the domain is small, this makes it difficult to add additional turbines without violating

430   the spacing constraint. Doing so would require adjusting the location of multiple turbines at once to make room, which is not something the algorithms can do.

The genetic algorithm with the gridded plant domain performs very well for each objective function. It even outperforms the gradient-based optimization for the COE objective and performs within 3% of the best solution found for the AEP and profit objectives. However, the computational expense was high, requiring the most time of any of the gradient-free algorithms and by

435   far the most function calls. However, because this problem is relatively small, the computational expense was not prohibitive.

The boundary-grid optimization solved with a genetic algorithm performs in the middle of the pack for the gradient-free algorithms. It performs very well with the AEP objective, but poorly with the COE and profit objective. This is because of the small wind plant area. The boundary-grid formulation forces turbines to be equally spaced around the boundary. With a unidirectional wind rose, this means that some turbines will always be in the back of the power plant relative to the incoming wind.

440   As discussed before, waked turbines were very detrimental for COE and, by extension, detrimental from a profit perspective as well.

Finally, the gradient-based optimizer, while sweeping across the number of wind turbines, performs well for each objective function. Because this algorithm allows the most freedom, permitting each turbine to be placed wherever the optimizer deems best, we always expected the gradient-based optimizations to perform well. In fact, we expected this optimizer to perform

445   the best of all for each objective, making it quite surprising that it is outperformed for both the AEP and COE objectives. Even though the gradient-free algorithms did not provide as much freedom, they were able to find the best solution for these objectives.

At this point, we would like to reiterate that the results shown in Table 2 are for a limited number of starting optimizations. They do not indicate that the solutions found are the best solution that each optimizer is capable of finding. These results simply

450   show the optimizer performance with a small sample size. That said, for a small number of discretizations, finding the optimal number and layout of wind turbines for various objectives can be achieved with a simple genetic algorithm.

**6.3 Large Power Plant with Unidirectional Wind Rose**

The performance and required computational expense of optimization algorithms can vary dramatically depending on the problem size. In this section, we will present another set of optimizations we ran for a larger domain. This wind power plant is

455   a square with 1.6-km sides. For the gridded domain, the plant is divided into a 20-by-20 square grid, which maintains the same spacing between grid points as in the small wind plant example. For a wind plant of this size, we did not run the gradient-based optimizer because of the large computational expense required to run the sweep across all of the possible optimal number of turbines. We only ran the optimizations for the four gradient-free methods we previously discussed. As in the previous section, we ran each optimization method 5 times to convergence for each objective. The wind resource and PPA for this wind power

460   plant were also assumed to be the same as for the small wind plant. The results for this wind power plant optimization are presented similarly to those for the small wind plant, with full results shown in Table 3, and the best layout for each objective shown in Fig. 8.

[Figure]

**Table 3.** Complete optimization results for the large wind power plant with unidirectional wind rose.

| objective | optimization | num. turbines/ avg. spacing (D) | optimal value/ normalized | time (s) | function calls |
|---|---|---|---|---|---|
| AEP (GWh) | greedy grid | **40**/2.55 | **663**/0.920 | 746 | 34,012 |
| | genetic grid | **23**/3.58 | **450**/0.624 | 2,539 | 44,095 |
| | sweep grid | **47**/2.32 | **710**/0.985 | 113 | 1,220 |
| | genetic BG | **48**/2.29 | **721**/1.000 | 3,310 | 48,824 |
| COE ($/MWh) | greedy grid | **22**/3.68 | **20.51**/1.007 | 423 | 28,393 |
| | genetic grid | **16**/4.53 | **21.00**/1.031 | 2,656 | 63,104 |
| | sweep grid | **20**/3.91 | **20.37**/1.000 | 119 | 4,895 |
| | genetic BG | **18**/4.19 | **20.73**/1.018 | 1,002 | 35,814 |
| annual profit ($MM) | greedy grid | **31**/2.97 | **4.98**/0.979 | 592 | 31,902 |
| | genetic grid | **20**/3.91 | **3.33**/0.655 | 2,826 | 47,977 |
| | sweep grid | **31**/2.97 | **4.93**/0.970 | 116 | 2,528 |
| | genetic BG | **33**/2.86 | **5.09**/1.000 | 2,126 | 52,869 |

**6.3.1 Large Power Plant with Unidirectional Wind Rose: Different Objectives**

The general trends that we observed from the smaller wind power plant optimizations hold true for this larger wind plant as
465 well. The wind plant optimized for AEP has as many wind turbines in the plant as possible without violating the spacing constraints, leading to a large number of turbines. The wind plant optimized for COE has turbines that are minimally waked, leading to a very small number of turbines. The wind plant optimized for profit is a middle ground between the previous two objectives. One main difference between the results for this large wind plant and the small wind plant is in the optimal solution for minimum COE. For the large wind plant, some of the turbines in the layout optimized for COE are more waked than in the
470 small wind plant. However, these waked turbines are far downstream of the waking turbine, meaning that the wind speed in the wake has recovered much of the way back to the freestream wind speed. In the small wind plant, there was not much space for the wakes to recover, meaning any amount of waking on a turbine was more detrimental to COE.

Figure 9 shows several metrics for the three wind power plants that were optimal for the different objective functions. Although the trends in this figure are similar to those in Fig. 7, the differences in metrics are more extreme for the different
475 objective functions. While all of the metrics are interesting, there are a few specific observations that are worth pointing out. First, the COE for the wind plant optimized for profit is relatively low, which is impressive because COE was never directly minimized. Second, the profit for the wind plant optimized for AEP is very low. Even though the plant produces a lot of energy, the tight turbine spacing causes high wake losses and inefficient turbines; thus, the high costs are not offset by the high energy production. Third, the wake losses for the wind plant optimized for minimum COE are very low, less than 2%. This is
480 impressively low and is only possible because of the unidirectional wind rose.

[Figure]

[Figure]

**Figure 8.** The optimal layouts for each objective for the large wind power plant with a unidirectional wind rose. From top to bottom, the associated objective functions are: AEP, COE, and profit. The text within each figure provides the value of all three metrics for each wind plant.

**6.3.2 Large Power Plant with Unidirectional Wind Rose: Different Algorithms**

While the general trends found for the solutions with different objectives were similar between the small and large wind power plants, the computational expense and performance of different algorithms were not. In the last three columns of Table 3, we see how well each optimization method performs. The most glaring difference is seen in the performance of the genetic algorithm
485 with the gridded turbine domain. While with the small wind plant, this method provided the best or near-best results for each

[Figure]

[Figure]

[Figure]

**Figure 9.** A comparison of the performance metrics of the three different wind power plants optimized for different objective functions. These results are shown for the large wind plant with a unidirectional wind rose.

objective, in this larger domain it severely underperformed compared to the other optimization methods. While the genetic algorithm was easily able to handle the 100 design variables from the 10-by-10 grid of the small wind plant, it appears unsuited to find a good solution for the 20-by-20 grid of the large wind plant. This observation agrees with our previous intuition with genetic algorithms in that they tend to perform poorly as problem complexity increases.

490 Next, the greedy algorithm and the repeated sweep algorithm both perform relatively better in the larger wind plant than they did in the smaller wind plant. For the AEP and profit objectives, these algorithms do not perform the best, but produce respectable results. For the COE objective, there are three different algorithms that provide basically the same result with optimal solutions within 1% of each other.

 Finally, the boundary-grid algorithm performs excellently for each objective function for this large wind power plant op-
495 timization. The organized structure forced by the boundary-grid method is able to fully take advantage of the unidirectional wind rose and create layouts with the appropriate number of turbines where waking is minimal. Even though it uses a genetic algorithm, the boundary-grid optimization performance does not suffer with the increased size of the wind plant. With this formulation, the number of design variables remains constant, independent of the number of turbines being modelled and the size of the domain.

500 **6.4 Large Power Plant with Full Wind Rose**

The final scenario in which we will examine the performance of each optimization algorithm is for the large wind power plant domain. Unlike the previous examples, the optimization results shown in this section include a full wind rose, discretized into 72 wind direction bins and shown in Fig. 10. The wind speeds are still assumed to be uniform at 10 m/s from each wind direction. Everything else for this optimization scenario is 
[revised manuscript text omitted]
 is not always the case; for most other cases, the boundary-grid method will perform comparatively well. The primary reason that the boundary-grid method is so effective is that the required function calls do not change with the problem size. While the greedy and repeated-sweep algorithms require many more function calls as the wind plant domain increases in size, the boundary-grid method remained constant between the small and large wind plants. At some point, the boundary-grid method will be more computationally efficient than any of the other algorithms because the number of design variables always remains constant.

**6.6 Varied Power Purchase Agreement**

[revised manuscript text omitted]

Although there are many sites in the United States that are capacity constrained (limited by the capacity of the transmission interconnection), the wind plants optimized for this paper assume land-constrained sites, which are also found throughout the United States. Examples of land-constrained sites in the United States with high turbine density are New England and San Gorgonio, California. The high turbine densities are driven by high PPAs in New England and exceptional wind resources and high-capacity transmission in San Gorgonio. Projects in New England are typically built on ridgelines or other terrain features that limit siting, while those in the San Gorgonio Pass have limited land availability because of terrain and the concentrated wind resource. Many of these wind plants have capacity densities between 5 MW/km$^2$ and 10 MW/km$^2$ compared to regional averages in the interior United States and the Great Lakes Region between 1.5 MW/km$^2$ and 3 MW/km$^2$.

Many of the optimal solutions presented in this paper place wind turbines very close together. As turbine scale and cost, energy prices, incentives, social considerations, land availability, resource, and transmission availability vary in the future, the power density of wind power plants in the United States will vary as well. Increasing turbine capacity while maintaining a constant specific power has been shown in some cases to increase potential capacity density. This can include setbacks from structures and other infrastructure as a function of tip height and constant minimum relative rotor spacing between turbines (Bons et al., 2019). Innovations such as wake steering may allow siting flexibility and may increase power densities and increase energy capture.

**7   Conclusions**

In this paper, we presented our research on wind power plant layout optimization, including optimizing the number of wind turbines. We specifically discussed the effect of different objective functions on the optimal solution as well as the pros and cons associated with using different problem formulations and optimizers to solve the problem. We explored optimizing several different wind plants for objectives of AEP, COE, and profit. The number of turbines in each optimal solution varied dramatically. The plants optimized for AEP tended to have the most wind turbines, those optimized for COE had the least, and those optimized for profit were somewhere in between. The purpose of this research is not to provide an optimal wind plant layout for a specific wind plant boundary and wind conditions, nor is it to suggest general rules of thumb for designing wind plants with different objective functions. The purpose is to clearly demonstrate that the solution from optimizing a wind plant can be heavily influenced by the objective function, particularly when considering the number of wind turbines as a design variable. Specific solutions and layouts will be determined by the models and problem parameters, but the objective must be carefully

660 chosen to represent the desired outcome because any mathematical optimizer will exploit this objective. From the models we used in this paper, the optimal number of turbines for a square wind plant with a full wind rose was 15, 24, or 54, depending on the objective of COE, profit, or AEP, respectively.

The other area that we discussed in this paper is problem formulation and algorithm selection for performing the optimization of turbine number and wind power plant layout optimization. We also presented a very simple repeated-sweep algorithm that

[revised manuscript text omitted]

---

## Author Comment (AC1)

**Response to Reviewer 1**

Andrew P. J. Stanley, Owen Roberts, Jennifer King, and Christopher J. Bay

May 2021

First, we would like the express our gratitude for your review of our paper. We realize that you took time out of a busy schedule to read this manuscript and provide feedback, for which we are very grateful. We have structured this response to be clear and easy to follow. Each of your original comments will be shown in blue, immediately followed by our response in black.

**1   Specific comments:**

**Chapter 1 (Introduction):**

Directly in the second sentence, the authors state that "wind turbines ... require no external fuel, and require little to no water". With the last statement, the reviewer guesses that the authors refer to the manufacturing of wind turbines, however, this on the other hand requires external fuels. Thus, the authors should be more precise: Either all statements in this sentence should only refer to the wind turbine during operation or more details on the separate statements should be provided.

This sentence was modified, it now reads: "Wind energy provides several advantages to the sustainable energy grid of the future. Wind turbines produce minimal carbon dioxide or other air pollution, require no external fuel during operation, and require little water throughout their lifetime."
* * *
References to relevant literature are missing in the third paragraph. Of course, most references are addressed in the cited review paper, however, some relevant literature should be added for each aspect outlined in this paragraph.

Several citations were added to each of the different optimization methods in this paragraph.
* * *
References are missing in the last paragraph. It is only referred to previous papers. Please be more precise what has already been done in other research work and what is novel in this study.

Several citations were added in this paragraph referencing previous work.
* * *
**Chapter 2 (Wake Model):**

Which coordinate system has been used? This does not become clear when the location (x, y, z) is introduced after Equation 2.

This entire section was reworked. The coordinate definition is now included immediately following what is now Equation 1, where $x$, $y$, and $z$ are first mentioned. The updated text now reads: "...the velocity at a desired location $(x, y, z)$, where $x$, $y$, and $z$ refer to the streamwise, cross-stream, and vertical coordinates, respectively ..."
* * *
*Just after Equation 4 the lateral and vertical direction are introduced as corresponding to y- and z-direction. This information would be required already earlier, directly after Equation 2. When mentioning this definition directly at the beginning, the question regarding the coordinate system might no longer be so relevant.*

Refer to the response from the previous comment.
* * *
*How are the nine locations (for determining the average rotor wind speed) distributed with respect to the lateral and vertical? Maybe a small figure could help explaining this.*

Good question and great idea, a new figure (now Figure 1) was added visualizing these points.
* * *
**Chapter 5 (Optimization Algorithms):**

*As the parameter for the maximum number of generations is used in the algorithm code, the specified number for the maximum number of generations should be addressed in the text (Section 5.2).*

The text was modified to include a description of the maximum generations, which now reads: "Convergence was assumed after the best performance was within a tolerance of $10^{-3}$ for 25 generations, or a maximum generation limit of 1000 was met. For the results in this paper, the maximum generation limit was never met."
* * *
*Why is at the end of Section 5.4 a range of 2-18 turbines defined, whereas in Section 5.5 the spacing constraint is only defined as two times the rotor diameter?*

The range 2-18 turbines refers to the number of turbines swept across in the gradient-based optimization, **not** the turbine spacing. Gradient-based optimization does not allow for discrete design variables, such as the number of turbines, so we had to try the optimization with each potential number of wind turbines to see which gave the best answer.
* * *
*Why are just 2D specified as minimum spacing? The authors themselves state later in Section 6.4.1 that this is not a realistic assumption. To work with more realistic example cases, why is not directly a spacing of for example the mentioned 5D used?*

We added the following sentence to the section describing the constraints to justify this decision: "This minimum spacing is on the small side, and is used to exaggerate the differences in the optimal solutions obtained with different objective function."
* * *
**Chapter 6 (Results):**

*Within the 1D example (Section 6.1) it is interesting to see that \$70/MWh and \$90/MWh yield the same optimum of 18 turbines. Maybe it can be further discussed on this result in the text as well.*

We thought so to, and think that is a great idea to add discussion on this topic. The following text was added to this section: "The optimal number of turbines increases from 13 to 16 as the PPA increases from \$30/MWh to \$50/MWh, then again to 18 as the PPA increases to \$70/MWh. However, when the PPA increases to \$90/MWh, the optimal number of turbines remains at 18. This is because the number of turbines is not continuous, and is only represented by integer values. For a given scenario, different PPA thresholds could be defined, above which the optimal number of turbines would increase by one. From Fig. 6, it appears that the optimal number of turbines is more sensitive at low PPA values, and becomes less sensitive as PPA increases. "
* * *
Why was the specific value of $30/MWh taken for the further example cases (mentioned at first in the second paragraph of Section 6.2)? In Section 6.6 and Figure 15 later in the paper, the authors present already a good reasoning for going for $30/MWh. This should be mentioned already (and as well) at this point.

Justification for this decision was added into the second paragraph of Section 6.2: "The PPA was assumed to be $30/MWh, which close to the COE solutions that were achieved, and is within the range of the PPAs of real wind farms from the past few years (see Fig. 16)."
* * *
Some statements in the discussions on the results are too generic. Thus, it is not really true that "the greedy and repeated sweep algorithms do not perform very well", as the repeated sweep algorithm finds the second-best solution for the COE objective (Section 6.2.2).

This paragraph was rewritten to two paragraphs, explaining the greedy and repeated sweep algorithms separately. The new paragraphs read:

"For this small wind power plant with the unidirectional wind rose, the greedy algorithm did not perform very well. If found the worst solution for both the AEP and COE objectives, and only the third best solution for the profit objective, but still underperformed by more than 8% compared to the best solution in this objective. This algorithm relies on placing turbines far apart to get the maximum benefit possible at each step of the optimization. Because the domain for this scenario was small, this made it difficult to add additional turbines without violating the spacing constraint. Doing so would require adjusting the location of multiple turbines at once to make room, which is not something this algorithm does. Although the computational expense for the greedy algorithm in this scenario was minimal, its poor performance does not justify its use.

For the objectives with higher turbine density, AEP and profit, the repeated sweep algorithm performed poorly. The answers for these objectives were either the worst or second worst solution found. However, for the COE objective this algorithm performed quite well and found the second best solution, within 0.2% of the best solution. Like the greedy algorithm, the repeated sweep algorithm has a step that relies on greedily placing turbines in the domain if they result in an improvement of the objective function. This algorithm has difficulty placing the turbines without violating spacing constraints for objectives that have many turbines in the optimal solution. For the COE objective however, the optimal number of turbines was much fewer. Thus the repeated sweep algorithm could place the turbines and move them around to a certain extent to find an excellent solution for this objective. The negligible computational expense for this algorithm could justify its use for this scenario, for an objective with optimal turbine spacings that are sufficiently larger than the minimum spacing constraints."
* * *
The sentence "For the large wind plant, some of the turbines in the layout optimized for COE are more waked than in the small wind plant" in the first paragraph of Section 6.3.1 needs some more explanations. Based on the Figures presenting the optimal layouts, as well as based on the presented numbers for the wake losses, the reader gets rather a different impression opposite to the statement in the above-mentioned sentence.

Thank you for pointing this out, this discussion was based on a previous result and was an error. We have updated the text explaining instead the optimal profit solution, which now reads:

"One main difference between the results for this large wind plant and the small wind plant is in the optimal solution for maximum profit. For the large plant, the wake losses for the optimal profit solution are significantly higher than for the small wind plant, 12% compared to 5.4% (see Figs. 10 and 8). More turbines can fit within the boundary of the large plant, pushing the plant further down the BOS cost curve shown in Fig. 3. The reduced costs from economies of scale make up for higher wake losses in the optimal solution."
* * *
Additional discussion was added to this section focusing on the differences in computational expense between the small and large plant scenarios. We also separated out the discussion of the greedy and repeated sweep algorithms to talk about each one individually.
* * *
What is the probability of occurrence taken for each wind direction bin in Section 6.4?

The probability of occurrence is shown with the windrose in what is now Figure 11. The text in the first paragraph of section 6.4.1 was rearranged to make this more apparent.
* * *
Especially with respect to the three different objectives, a case with realistic annual wind distribution would be interesting to be investigated and of most relevant meaning for real applications.

We fully agree. More wind resource distributions (of direction probability and speed) would be interesting to examine in this context. We have also thought about different/more realistic wind farm boundaries, perhaps with different discrete sections available for development, coupled optimization of turbine design and layout for different objectives, and co-located wind and solar plants. There is so much that can be done in this space with future work!
* * *
The discussion and especially last statement ("However, the optimizer didn't find this solution from the five optimizations that we ran") in the first paragraph of Section 6.4.1 are not satisfactory. There is a significant difference between the results of the two cases (48 versus 54) and furthermore there is a large discrepancy between the results from the different optimizers, presented in Table 4. Furthermore, the last sentence leaves the question on the sensitivity and trustfulness of the results, as the results cannot be repeated and there is some random chance to score one time maybe better or not or just another time.

The following text was added to the end of this paragraph addressing the difference you point out:

"Setting up an optimization run always involves a trade-off between trying to find the best solution and minimizing computational expense. One can imagine two extremes for a genetic algorithm. The first extreme has an enormous population size and very strict convergence criteria. This optimization would theoretically find a very good, maybe the best solution, but at a restrictively high computational expense. The other extreme would have a minuscule population and very lax convergence criteria. This population would converge very quickly, but would lend very little confidence that a good solution was found. For this paper, our goal was to examine overall trends, and not to find the global solution for every scenario and optimizer combination. For a one-off optimization, it may be prudent to run more than 5 optimizations with different initialization of the design space, and maybe tune the optimizer parameters and convergence criteria to the specific problem. However, for this paper our goal was to run a large quantity of optimizations across a range of scenarios, which required us to make some decisions to keep the computational expense reasonable. Even though the optimizers for the large wind farm and unidirectional wind rose scenario failed to find the best solution for the AEP objective, we can stand behind our methodology and have confidence that the general trends we have observed are accurate and valid."
* * *
Section 6.7 could go into a separate new Chapter. In general, the reviewer suggests to have another Discussion Chapter (before the final Chapter Conclusions), in which additionally the assumptions made and considered example cases should be investigated in more detail, e.g. with respect to realistic cases (see some comments before on the wind speed distribution or the spacing constraints), required further sensitivity studies, meaningfulness of the results due to the specified optimization settings (limited number of iterations, no repeatability), . . .

Section 6.7 was split out into it's own chapter, and another section was added that addresses these points.

**2 Technical corrections:**

Chapter 1 (Introduction): A separate paragraph at the end of Chapter 1 (Introduction), in which the structure of the paper is presented, would be useful to help navigating the reader through the paper.

This paragraph was added to the end of the Introduction: "The rest of this paper is outlined as follows: Sec. 2 presents the wake model we used in this paper, and the relevant turbine parameters, Sec. 3 presents the power models, cost models, and how they are combined to form the 3 objective functions we explored in this paper, Sec. 4 describes the different sets of design variables we used to define the locations of wind turbines, Sec. 5 explains the optimization algorithms we used in this paper, Sec. 6 presents and discusses the results from our optimizations, and Sec. 7 contains our conclusions from this work."
* * *
Some word repetitions should be avoided (e.g. addressed in the fourth paragraph of Chapter 1).

We went through the paper specifically looking out for and fixing excessive word repetitions.
* * *
Chapter 2 (Wake Model): The definition of the parameter I0 might not be relevant, as this is not used in the equations.

This entire section was reworked, now $I$ is used and defined.
* * *
Figures and tables should be placed in such a way that text within a paragraph is not separated if this is not required (e.g. Table 1 and Figure 1 or Figure 2).

This issue will be taken care of when the paper is typeset before final publication.
* * *
The Greek symbol should be used as well in the text instead of writing phi (Section 3.1).

This change was incorporated.
* * *
Throughout the paper it should be ensured that the tenses are used consistently.

We went through the paper to to fix inconsistent tenses.
* * *
Chapter 6, first sentence: the word "of" is missing between results and our.

This change was incorporated.
* * *
Figure 5: Please complete the legend in the right plot, such as "PPA in \$/MWh".

This change was incorporated.
* * *
The reference to "previous section" in the second sentence of Section 6.6 is wrong. Please use the reference to the number of the specific section (here to 6.4) to be clear.

Except for one occurrence that was talking about previous sections in general, all mentions of "previous section" throughout the text were replaced with the specific section number they are referring to.

---

## Author Comment (AC2)

**Response to Reviewer 2**

Andrew P. J. Stanley, Owen Roberts, Jennifer King, and Christopher J. Bay

May 2021

First, we would like the express our sincere thanks for reviewing our paper. We know it is a time consuming process, and we are extremely grateful for your thorough and thoughtful review. We have structured this response to be clear and easy to follow. Each of your original comments will be shown in red, immediately followed by our response in black. Keep in mind that the line numbers associated with each comment correspond to the original draft. After our changes, there will be some offset between the original line numbers and those in the updated manuscript.

**1 Structure:**

The paper has a potentially high scientific impact and while showing high quality in its results. However, although it is well structured in general, the results on the performance of the different algorithms would become more fluent and dynamic if the sections for each case study (6.2.2, 6.3.2, 6.4.2) were concentrated only in section 6.5, a section by the way already summarizing the performance of the algorithms on all cases.

Thank you! And thank you for your thoughts on the paper organization. The purpose of Sections 6.2.2, 6.3.2, and 6.4.2 is to discuss the differences in the algorithm performance **within each specific problem setup**. On the other hand, Section 6.5 discusses the overall performance of each algorithm across the entire range of problems tested in this paper. Even though there is some repetition of information, we feel that is is helpful to discuss our results from these different perspectives, and keep the discussions of the algorithm performances within each section they are referring to. We have not made any changes to the text based on this comment.

**2 Specific comments:**

Line 22: crossed out "Because of economies of scale"

This was reworded to say: "Because of economies of scale, utility-scale wind turbines are deployed in groups."
* * *
Line 62: you cannot talk about results in the introduction. You have to talk in general terms from what literature says or ellaborate a general rationale.

We feel that providing a small amount of specific results from our paper is helpful for the reader's experience, and gives a preview of one of the important takeaways of our paper. After consideration, we have decided to keep text from the original manuscript.
* * *
Line 67: See Li, Wenwen, Ender Özcan, and Robert John. "Multi-objective evolutionary algorithms and hyper-heuristics for wind farm layout optimisation." Renewable Energy 105 (2017): 473-482. There, they compare the impact of 2 different COE definitions on the number of turbines (Fig. 2).

and

Line 68: Again, argument with some rationale or literature.

The mentioned citation was added, and the text was reworded to say:

"Li et al. began to explore this sensitivity with multi-objective optimization of wind farm layout and turbine number,considering AEP and COE. As part of their paper, these authors studied how different formulations of the COE definition affect the final solutions (Li et al., 2017). Balasubramanian et al. also mention the importance of appropriately defining the objective for wind farm layout optimization (Balasubramanian et al., 2020). In our paper, we include an empirically based cost model, and explore three different objectives in our single objective optimization formulation to further understand the sensitivity of wind farm layout and turbine number to the objective."
* * *
Line 69: Please include some state-of-the-art dissertation

This was reworded to say: "In past research on wind farm layout optimization, there has been a wide variety of algorithms and problem formulations used, with little consensus on which strategies are the best (Shakoor et al., 2016; Baker et al., 2019; Hou et al., 2019; Balasubramanian et al., 2020)."
* * *
Line 89: what does delta actually represent? and which values are you using for it?

and

Line 91: 1) how is that dependency?? 2) which value do you use for I0 and how does it change for further downstream turbines??

and

Line 96: you have u_inf, but how do you calculate uR and u0? Please clarify/unify/simplify expressions.

and

Line 99: how do you define k? / which values do you assign to k?

Section 2 of the paper was reworked, with several clarifications and corrections to the original submission. See the revised section of the manuscript.
* * *
Line 107: In general for this section: You have to explain when a certain CF can be more convenient for some cases and which one for others.

We understand where you are coming from, thanks for this comment. However, the purpose of our paper is not to present one objective function or another as "correct" for different situations, but to demonstrate that the solution is sensitive to the objective function, indicating that correctly defining the objective of a wind farm layout optimization is important. It is up to those who are designing the wind farm to define an objective function that prioritizes what is most important to them. We have not made any changes to the manuscript from this comment.
* * *
Line 155: why?

This sentence was reworded to say: "Because a primary interest of most businesses is to make money, this objective would likely be of more interest to wind power plant developers, as opposed to AEP or COE previously discussed."

**Line 159: I don't see how would you be able to optimize with a non constant PPA.**

This was clarified. The new manuscript says: "For this paper, we assumed that the PPA was a constant, as opposed to using time of day pricing, seasonal or yearly PPA adjustments, or including PPA incentives or penalties for power quality. In a given optimization the PPA is defined as a constant, although we will vary this constant to study its effect during different optimizations."

**Line 178: Then why some works consider a full continuous search space?**

We're not exactly sure what you mean with this question, but we think you are asking "if computational expense increases with the number of design variables, how can some optimization methods treat the domain as continuous which would essentially be an infinite number of design variables?" The statement that you are referring to with this question is talking about the number of **design variables**, and not about whether they are formulated as continuous or discrete. In the grid formulation, the design space is split into a grid of Boolean design variables, which either contain or do not contain a turbine, meaning that the number of design variables increase with grid refinement. If the location of every turbine were to be defined with it's own continuous x and y coordinate design variable, then the number of variables would remain fixed with the number of turbines and would not suffer from this increase in computational expense with increasing design variables.

**Line 190: In this case (BG), will each grid point hold a turbine? or they might or might not hold it? Please clarify**

The following was added to clarify this point: "For the gridded domain design variables, the grid defined potential turbine locations which were assigned a Boolean value during the optimization to determine if they had a turbine. In the boundary-grid method, the design variables directly determine the location of every turbine in the farm, meaning there is always a turbine placed at the points defined by the boundary-grid parameterization."

**Line 198: could you show the role of (cx,cy) in the problem encoding? These coordinates look strongly redundant, as the rotation of the parallelogram can be defined just with theta.**

The entire grid moves uniformly with the center (x, y). The following was added to clarify this point: "The center of the grid is shown as the point $(cx, cy)$, which determines the translation of all points in the grid"

**Line 224: please specify the mechanism that adds/removes turbines in this algorithm**

Good question. There is no mechanism that is used to specifically add or remove turbines. In a discrete genetic algorithm, the design variables are encoded in binary strings. When the strings for each design variable are appended, the resultant string is called an "individual." The entire "population" is the set of individuals, each representing a different combination of design variables and corresponding to a certain objective function value. Each set of design variable can correspond to a different number of turbines. The genetic algorithm randomly initializes the population, then proceeds to combine individuals in different ways and randomly switch individual bits ("mutate"). Combinations and mutations that improve the objective function are kept, while the others are removed. This proceeds until convergence is met. For the gridded parameterization, the 0's and 1's defining each individual directly correspond to the number of turbines (the number of 1's is the number of turbines in the farm). However, for the boundary-grid parameterization the 0's and 1's map to each of the 11 design variables, which then determine the number of turbines. We have not added any further description in the text, as this information is considered general knowledge about genetic algorithms.

Line 227: what is your criteria when setting the parameters of the genetic algorithm? Justify

We added the following statement to the paper: "We chose the tuning parameters for our algorithm with a combination of trial and error and best practice recommendations."

Algorithm 2: please specify the mechanism that adds/removes turbines

Refer to the previous explanation about the genetic algorithm.

Line 241: Could you at least mention some reference or argument on which you base your method? Otherwise it looks like this algorithm arises from nowhere. If it is the result of a trial and error process, please specify.

The following text was added to give a small background into the creation of this algorithm:

"The creation of this optimizer was inspired by attempting to apply gradient-based optimization principles to discrete design variables. As described below, the algorithm works by comparing adjacent points, and switching the values if it would improve the objective function, which could be imagined as the discrete version of a gradient."

Line 270: sounds weird

This sentence was reworded to say: "For one case discussed in Sec. 6.2 we also used gradient-based optimization in order to compare the results."

Line 279: what does this stand for? is this some kind of sequential quadratic programming?

This was reworded to say: "For the results in this paper, we used the open-source SLSQP (Sequential Least Squares Programming) optimizer available in SciPy."

Line 287: of

This change was incorporated.

Line 298: very informal style

"This figure is particularly interesting when you notice that . . . " was changed to "One key takeaway from this figure is that . . . "

Line 339: This sounds strange placed here. Perhaps move to conclusions.

and

Line 344: I am completely lost here. You might want to say that smaller projects might be only feasible with a high PPA price..

and

Line 345: smaller plant sizes have higher construction costs??? COE might be higher, but not the overall project or construction costs... please modify

This paragraph was reworked. The new paragraph reads:

"Historically, capacity expansion models have assumed a constant power density that does not vary with the PPA. Not only does Fig. 6 demonstrate the differences between a minimum COE objective and a maximum profit objective, but it also shows that the cost modeling assumptions can greatly affect the optimal number of turbines in a given land-constrained wind plant. Aggressive carbon reduction scenarios or other renewable energy goals typically result in high PPAs for renewables, which would lead to a higher optimal number of turbines and higher capacity densities for land constrained sites. This has important implications for capacity expansion models and could play a role in the future deployment of wind, as capacity density may often be much higher than is currently assumed."
* * *
Table 2: do not use / unless you are dividing. it is confusing and an erroneous notation.

Tables 2—4 were modified to have these values in their own column
* * *
Line 388: By now we only know that this happens using a unidirectional wind rose. Move down or remove

We qualified the statement to say there is a potential of this behavior, rather than a guarantee. The updated sentence says: "If we were to repeat the optimization for a much larger domain we could potentially see results similar to Fig. 5, where having too many turbines could actually be detrimental for AEP."
* * *
Line 392: yes it exists. See Figure 7

This incorrect claim was removed. The new sentence reads: "The turbines are arranged such that waking is minimal."
* * *
Line 393: So far only in this case. Please emphasize.

"We can conclude that . . ." was changed to "For this case, we can conclude that . . ."
* * *
Line 400: A good question here would be which PPA value would be equivalent to apply a COE minimization and an AEP maximization. Could you derive that equivalence?

Good question! The short answer is no, we can't. A COE minimization and an AEP maximization would entail a multi-objective optimization, and would result in a set of solutions (a Pareto front) instead of a single point. It would be possible to find a PPA that results in the same solution for some of the individual points on the Pareto front, but that is beyond the scope of this paper.
* * *
Line 422: what are exactly function calls? you mean the evaluation via the wake model? Then please specify. On what depends the number of calls?

The following text was modified in the first mention of functions calls in the paper, which now reads ". . . shows total number of calls to the wind farm evaluation, or function calls, required to run . . ."

The number of function calls is an indication of the efficiency of the optimizer, and how many feasible solutions are tested by the optimizer. Runs that quickly converge on a solution and don't need to keep iterating will have relatively fewer function calls, and algorithms and problem formulations that tend to

generate infeasible combinations of design variables will have relatively fewer function calls because the objective function is not fully called for a solution that violates the constraints.
* * *
Line 435: When a wind energy project of thousands millions is about to be deployed, I don't think 1 week or even longer optimization times are not worth.

This thought is understandable, and is actually pretty common when discussing the computational expense of wind farm optimization. For a single run that is guaranteed to find a solution (which is not at all guaranteed), a time scale of a couple of weeks or maybe even months is not particularly problematic. However, there are other considerations that are important, including setting up the optimization problem, which requires several iterations and small adjustments, wanting to try several optimizations with different parameters such as different turbine models or cost model assumptions, higher fidelity models for which the computation time is much higher per function call, and research (such as this paper) where the turn around time is lower and more configurations are generally tested. For these reasons we can see that computational expense in wind farm optimization, including wind farm layout optimization, is important to consider.
* * *
Line 443: what???

This was reworded to say: "Because this algorithm uses continuous design variables and allows full access to the wind plant domain . . . "
* * *
Line 446: which ones?

This was clarified, the new text reads: "Even though the greedy, genetic, and repeated-sweep algorithms did not provide as much freedom as the gradient-based optimization . . . "
* * *
Line 448: Within gradient-free algorithms, If an algorithm is efficient, the different launches will tend to provide very similar results each other. The extreme contrary case would be a complete random search (which might even be a good solution in some problems). Hence, it would be interesting to see the dispersion in the final results.

This is a great point, this would be interesting to consider in future work.
* * *
Line 455: Highlighted but no note written

We are not sure what you meant with this highlight, but we clarified the statement a big further. The new text says: "For the gridded domain, the plant is divided into a 20-by-20 square grid which, because of the increased size of the plant area, maintains the same spacing between grid points as in the small wind plant example."
* * *
Table 4: Is this 44 hours? If so, please mention these kind of computation times in the text.

A new column was added to each of the results tables with the time expressed in hours, in addition to the seconds column.
* * *
Line 540: text very redundant. Summarize in 6.5

Refer back to our response to the first comment regarding this organization.
* * *
**Line 545: too informal**

This was replaced with "does not"
* * *
**Line 579: sounds strange**

"be absolutely clear" was replaced with "reiterate"
* * *
**Line 603: 2**

This change was incorporated.
* * *
**Line 617: Why (on average) PPA decreses over time?**

The following sentence was added at the end of Section 6.6 to answer this question: "This decrease in PPA prices in recent years has been the combined result of higher capacity factors, declining installation costs and operating costs, and low interest rates (Wiser and Bolinger, 2019)."
* * *
**Line 640: You cannot compare to them, because your wake model is assumed to work only on flat terrain or offshore. In complex terrain, orography and high roughness lengths produce high turbulence levels, which make wakes to dissipate more rapidly, this allowing bigger power densities. Please emphasize.**

The intent of this paragraph is simply to inform the reader that land-constrained sites do exist and are fairly common, not to compare our results to the specific farms mentioned. We removed part of this paragraph to remedy this.
* * *
**Line 642: you have any idea why?**

**and**

**Line 644: This is not clear. Please reword.**

We decided to remove this paragraph.
* * *
**Line 653: Suggestion: "The number of turbines in each optimal solution showed highly sensitive to the objective function used".**

This sentence was reworded to say: "The number of turbines in each optimal solution was highly sensitive to the objective function that was used."
* * *
**Line 667: you mean the 20-by-20 grid? because in line 455 you say it has the same spacing as the 10-by-10 grid**

This was reworded to say "However, for a 20-by-20 grid . . . "